# Decentralized Learning in Online Queuing Systems

**Flore Sentenac**[*]
CREST, ENSAE Paris, Palaiseau, France
flore.sentenac@gmail.com

**Etienne Boursier**[*]
Centre Borelli, ENS Paris-Saclay, France
etienne.boursier1@gmail.com

**Vianney Perchet**
CREST, ENSAE Paris, Palaiseau, France
CRITEO AI Lab, Paris, France
vianney.perchet@normalesup.org

## Abstract

Motivated by packet routing in computer networks and resource allocation in radio networks, online queuing systems are composed of queues receiving packets at different rates. Repeatedly, they send packets to servers, each of them treating only at most one packet at a time. In the centralized case, the number of accumulated packets remains bounded (i.e., the system is *stable*) as long as the ratio between service rates and arrival rates is larger than 1. In the decentralized case, individual no-regret strategies ensures stability when this ratio is larger than 2. Yet, myopically minimizing regret disregards the long term effects due to the carryover of packets to further rounds. On the other hand, minimizing long term costs leads to stable Nash equilibria as soon as the ratio exceeds $\frac{e}{e-1}$. Stability with decentralized learning strategies with a ratio below 2 was a major remaining question. We first argue that for ratios up to 2, cooperation is required for stability of learning strategies, as selfish minimization of policy regret, a *patient* notion of regret, might indeed still be unstable in this case. We therefore consider cooperative queues and propose the first learning decentralized algorithm guaranteeing stability of the system as long as the ratio of rates is larger than 1, thus reaching performances comparable to centralized strategies.

## 1 Introduction

Inefficient decisions in repeated games can stem from both strategic and learning considerations. First, strategic agents selfishly maximize their own individual reward at others' expense. The *price of anarchy* [Koutsoupias and Papadimitriou, 1999] measures this inefficiency as the social welfare ratio between the best possible situation and the worst Nash equilibrium. Although reaching the best collective outcome might be illusory for selfish agents, considering the worst Nash equilibrium might be too pessimistic. In games with external factors, more complex interactions intervene and might lead the agents to the best equilibrium. Instead, the *price of stability* [Schulz and Moses, 2003] measures the inefficiency by the social welfare ratio between the best possible situation and the best Nash equilibrium.

Second, the agents also have to learn their environment, by repeatedly experimenting different outcomes. Learning equilibria in repeated games is at the core of many problems in computer science and economics [Fudenberg et al., 1998, Cesa-Bianchi and Lugosi, 2006]. The interaction between multiple agents can indeed interfere in the learning process, potentially converging to no or bad equilibria. It is yet known that in repeated games, if all agents follow no internal regret strategies, their actions converge in average to the set of correlated equilibria [Hart and Mas-Colell, 2000, Blum and Monsour, 2007, Perchet, 2014].

---

[*]Equal contributions

35th Conference on Neural Information Processing Systems (NeurIPS 2021).

Many related results are known in the classical repeated games [see e.g., Cesa-Bianchi and Lugosi, 2006, Roughgarden, 2010], where a single game is repeated over independent rounds (but the agents strategies might evolve and depend on the history). Motivated by packet routing in computer networks, Gaitonde and Tardos [2020a] introduced a repeated game with a *carryover* feature: the outcome of a round does not only depend on the actions of the agents, but also on the previous rounds. They consider heterogeneous queues sending packets to servers. If several queues simultaneously send packets to the same server, only the oldest packet is treated by the server.

Because of this carryover effect, little is known about this type of game. In a first paper, Gaitonde and Tardos [2020a] proved that if queues follow suitable no-regret strategies, a ratio of 2 between server and arrival rates leads to stability of the system, meaning that the number of packets accumulated by each queue remains bounded. However, the assumption of regret minimization sort of reflects a myopic behavior and is not adapted to games with carryover. Gaitonde and Tardos [2020b] subsequently consider a patient game, where queues instead minimize their asymptotic number of accumulated packets. A ratio only larger than $\frac{e}{e-1}$ then guarantees the stability of the system, while a smaller ratio leads to inefficient Nash equilibria. As a consequence, going below the $\frac{e}{e-1}$ factor requires some level of cooperation between the queues. This result actually holds with perfect knowledge of the problem parameters and it remained even unknown whether decentralized learning strategies can be stable with a ratio below 2.

We first argue that decentralized queues need some level of cooperation to ensure stability with a ratio of rates below 2. Policy regret can indeed be seen as a patient alternative to the regret notion. Yet even minimizing the policy regret might lead to instability when this ratio is below 2. An explicit decentralized cooperative algorithm called ADEQUA (A DEcentralized QUeuing Algorithm) is thus proposed. It is the first decentralized learning algorithm guaranteeing stability when this ratio is only larger than 1. ADEQUA does not require communication between the queues, but uses synchronisation between them to accurately estimate the problem parameters and avoid interference when sending packets. Our main result is given by Theorem 1 below, whose formal version, Theorem 5 in Section 4, also provides bounds on the number of accumulated packets.

**Theorem 1** (Theorem 5, informal). *If the ratio between server rates and arrival rates is larger than 1 and all queues follow* ADEQUA*, the system is strongly stable.*

The remaining of the paper is organised as follows. The model and existing results are recalled in Section 2. Section 3 argues that cooperation is required to guarantee stability of learning strategies when the ratio of rates is below 2. ADEQUA is then presented in Section 4, along with insights for the proof of Theorem 1. Section 5 finally compares the behavior of ADEQUA with no-regret strategies on toy examples and empirically confirms the different known theoretical results.

## 1.1 Additional related work

Queuing theory includes applications in diverse areas such as computer science, engineering, operation research [Shortle et al., 2018]. Borodin et al. [1996] for example use the stability theorem of Pemantle and Rosenthal [1999], which was also used by Gaitonde and Tardos [2020b], to study the problem of packet routing through a network. Our setting is the single-hop particular instance of throughput maximization in wireless networks. Motivated by resource allocation in multihop radio problem, packets can be sent through more general routing paths in the original problem. Tassiulas and Ephremides [1990] proposed a first stable centralized algorithm, when the service rates are known *a priori*. Stable decentralized algorithms were later introduced in specific cases [Neely et al., 2008, Jiang and Walrand, 2009, Shah and Shin, 2012], when the rewards $X_k(t)$ are observed before deciding which server to send the packet. The main challenge then is coordination and queues aim at avoiding collisions with each other. The proposed algorithms are thus not adapted to our setting, where both coordination between queues and learning the service rates are required. We refer the reader to [Georgiadis et al., 2006] for an extended survey on resource allocation in wireless networks.

Krishnasamy et al. [2016] first considered online learning for such queuing systems model, in the simple case of a single queue. It is a particular instance of stochastic multi-armed bandits, a celebrated online learning model, where the agent repeatedly takes an action within a finite set and observes its associated reward. This model becomes intricate when considering multiple queues, as they interfere when choosing the same server. It is then related to the multiplayer bandits problem which considers

multiple players simultaneously pulling arms. When several of them pull the same arm, a *collision* occurs and they receive no reward [Anandkumar et al., 2010].

The collision model is here different as one of the players still gets a reward. It is thus even more closely related to competing bandits [Liu et al., 2020a,b], where arms have preferences over the players and only the most preferred player pulling the arm actually gets the reward. Arm preferences are here not fixed and instead depend on the packets' ages. While collisions can be used as communication tools between players in multiplayer bandits [Bistritz and Leshem, 2018, Boursier and Perchet, 2019, Mehrabian et al., 2020, Wang et al., 2020], this becomes harder with an asymmetric collision model as in competing bandits. However, some level of communication remains possible [Sankararaman et al., 2020, Basu et al., 2021]. In queuing systems, collisions are not only asymmetric, but depend on the age of the sent packets, making such solutions unsuited.

While multiplayer bandits literature considers cooperative players, Boursier and Perchet [2020] showed that cooperative algorithms could be made robust to selfish players. On the other hand, competing bandits consider strategic players and arms as the goal is to reach a bipartite stable matching between them. Despite being cooperative, ADEQUA also has strategic considerations as the queues' strategy converges to a correlated equilibrium of the patient game described in Section 2.

An additional difficulty here appears as queues are asynchronous: they are not active at each round, but only when having packets left. This is different from the classical notion of asynchronicity [Bonnefoi et al., 2017], where players are active at each round with some fixed probability. Most strategies in multiplayer bandits rely on synchronisation between the players [Boursier and Perchet, 2019] to avoid collisions. While such a level of synchronisation is not possible here, some lower level of synchronisation is still used to avoid collisions between queues.

## 2 Queuing Model

We consider a queuing system composed of $N$ queues and $K$ servers, associated with vectors of arrival and service rates $\boldsymbol{\lambda}, \boldsymbol{\mu}$, where at each time step $t = 1, 2, \ldots$, the following happens:

- each queue $i \in [N]$ receives a new packet with probability $\lambda_i \in [0, 1]$, that is marked with the timestamp of its arrival time. If the queue currently has packet(s) on hold, it sends one of them to a chosen server $j$ based on its past observations.

- Each server $j \in [K]$ attempts to clear the oldest packet it has received, breaking ties uniformly at random. It succeeds with probability $\mu_j \in [0, 1]$ and otherwise sends it back to its original queue, as well as all other unprocessed packets.

At each time step, a queue only observes whether or not the packet sent (if any) is cleared by the server. We note $Q_t^i$ the number of packets in queue $i$ at time $t$. Given a packet-sending dynamics, the system is **stable** if, for each $i$ in $[N]$, $Q_t^i/t$ converges to 0 almost surely. It is **strongly stable**, if for any $r, t \geq 0$ and $i \in [N]$, $\mathbb{E}[(Q_t^i)^r] \leq C_r$, where $C_r$ is an arbitrarily large constant, depending on $r$ but not $t$. Without ambiguity, we also say the policy or the queues are (strongly) stable. Naturally, a strongly stable system is also stable [Gaitonde and Tardos, 2020a].

In the following, $x_{(i)}$ will denote the $i$-th order statistics of a vector $\boldsymbol{x}$, i.e., $\lambda_{(1)} \geq \lambda_{(2)} \geq \ldots \geq \lambda_{(N)}$ and $\mu_{(1)} \geq \ldots \geq \mu_{(K)}$. Without loss of generality, we assume $K \geq N$ (otherwise, we simply add fictitious servers with 0 service rate). The key quantity of a system is its **slack**, defined as the largest real number $\eta$ such that:

$$\sum_{i=1}^{k} \mu_{(i)} \geq \eta \sum_{i=1}^{k} \lambda_{(i)}, \ \forall \, k \leq N.$$

We also denote by $\mathcal{P}([K])$ the set of probability distributions on $[K]$ and by $\Delta$ the **margin** of the system defined by

$$\Delta := \min_{k \in [N]} \frac{1}{k} \sum_{i=1}^{k} (\mu_{(i)} - \lambda_{(i)}). \tag{1}$$

Notice that the alternative system where $\tilde{\lambda}_i = \lambda_i + \Delta$ and $\tilde{\mu}_k = \mu_k$ has a slack 1. In that sense, $\Delta$ is the largest *margin* between service and arrival rates that all queues can individually get in the system. Note that if $\eta > 1$, then $\Delta > 0$. We now recall existing results for this problem, summarized in Figure 1 below.

**Theorem 2** (Marshall et al. 1979). *For any instance, there exists a strongly stable centralized policy if and only if $\eta > 1$.*

**Theorem 3** (Gaitonde and Tardos 2020a, informal). *If $\eta > 2$, queues following appropriate no regret strategies are strongly stable.*
*For each $N > 0$, there exists a system and a dynamic s.t. $\eta > 2 - o(1/N)$, all queues follow appropriate no-regret strategies, but they are not strongly stable.*

In the above theorem, an *appropriate no regret strategy* is a strategy such that there exists a partitioning of the time into successive windows, for which the incurred regret is $o(w)$ with high probability on any window of length $w$. This for example includes the EXP3.P.1 algorithm [Auer et al., 2002] where the $k$-th window has length $2^k$.

The patient queuing game $\mathcal{G} = ([N], (c_i)_{i=1}^n, \boldsymbol{\mu}, \boldsymbol{\lambda})$ is defined as follows. The strategy space for each queue is $\mathcal{P}([K])$. Let $\boldsymbol{p}_{-i} \in (\mathcal{P}([K]))^{N-1}$ denote the vector of fixed distributions for all queues over servers, except for queue $i$. The cost function for queue $i$ is defined as:

$$c_i(p_i, \boldsymbol{p}_{-i}) = \lim_{t \to +\infty} T_t^i / t,$$

where $T_t^i$ is the age of the oldest packet in queue $i$ at time $t$. Bounding $T_t^i$ is equivalent to bounding $Q_t^i$.

**Theorem 4** (Gaitonde and Tardos 2020b, informal). *If $\eta > \frac{e}{e-1}$, any Nash equilibrium of the patient game $\mathcal{G}$ is stable.*

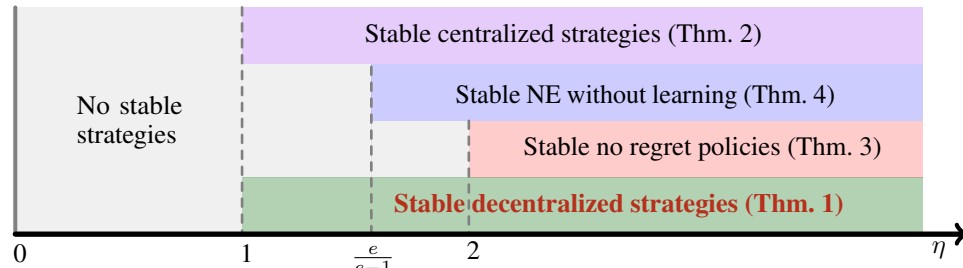

Figure 1: Existing results depending on the slack $\eta$. Our result is highlighted in red.

## 3  The case for a cooperative algorithm

According to Theorems 3 and 4, queues that are patient enough and select a fixed randomization over the servers are stable over a larger range of slack $\eta$ than queues optimizing their individual regret. A key difference between the two settings is that when minimizing their regret, queues are myopic, which is formalized as follows. Let $a_{1:t} = (a_1, ..., a_t)$ be the vector of actions played by a queue up to time $t$ and let $\nu_t(a_{1:t})$ be the indicator that it cleared a packet at iteration $t$. Classical (external) regret of queue $i$ over horizon $T$ is then defined as:

$$R_i^{\text{ext}}(T) := \max_{p \in \mathcal{P}([K])} \sum_{t=1}^{T} \mathbb{E}_{\tilde{a}_t \sim p}[\nu_t(a_{1:t-1}, \tilde{a}_t)] - \sum_{t=1}^{T} \nu_t(a_{1:t}).$$

Thus minimizing the external regret is equivalent to maximizing the instant rewards at each iteration, ignoring the consequences of the played action on the state of the system. However, in the context of queuing systems, the actions played by the queues change the state of the system. Notably, letting other queues clear packets can be in the best interest of a queue, as it may give it priority in the subsequent iterations where it holds older packets. Since the objective is to maximize the total number of packets cleared, it seems adapted to minimize a *patient* version of the regret, namely the policy regret [Arora et al., 2012], rather than the external regret, which is defined by

$$R_i^{\text{pol}}(T) := \max_{p \in \mathcal{P}([K])} \sum_{t=1}^{T} \mathbb{E}_{\tilde{a}_{1:t} \sim \otimes_{i=1}^t p}[\nu_t(\tilde{a}_{1:t})] - \sum_{t=1}^{T} \nu_t(a_{1:t}).$$

That is, $R_i^{\text{pol}}(T)$ is the expected difference between the number of packets queue $i$ cleared and the number of packets it would have cleared over the whole period by playing a fixed (possibly random) action, taking into account how this change of policy would affect the state of the system.

However, as stated in Proposition 1, optimizing this patient version of the regret rather than the myopic one could not guarantee stability on a wider range of slack value. This suggests that adding only patience to the learning strategy of the queues is not enough to go beyond a slack of 2, and that any strategy beating that factor 2 must somewhat include synchronisation between the queues.

**Proposition 1.** *Consider the partition of the time $t = 1, 2, \ldots$ into successive windows, where $w_k = k^2$ is the length of the $k$-th one. For any $N \geq 2$, there exists an instance with $2N$ queues and servers, with slack $\eta = 2 - \mathcal{O}\left(\frac{1}{N}\right)$, s.t., almost surely, each queue's policy regret is $o\left(w_k\right)$ on all but finitely many of the windows, but the system is not strongly stable.*

To ease comparison, the formulation in the above proposition matches that of the counter-example used to prove Theorem 3 (Gaitonde and Tardos 2020a). In that counter-example, a set of system parameters, for which *any* no external regret policies were unstable, was exhibited. Whereas we exhibit in our case a *specific* strategy that satisfies the no policy regret condition, but is unstable.

*Sketch of proof.* Consider a system with $2N$ queues and servers with $\lambda_i = 1/2N$ and $\mu_i = 1/N - 1/4N^2$ for all $i \in [2N]$. The considered strategy profile is the following. For each $k \geq 0$, the $k^{\text{th}}$ time window is split into two stages. During the first stage, of length $\lceil \alpha w_k \rceil$, queues $2i$ and $2i + 1$ both play server $2i + t \pmod{2N}$ at iteration $t$, for all $i \in [N]$. During the second stage of the time window, queue $i$ plays server $i + t \pmod{2N}$ at iteration $t$. This counter example, albeit very specific, illustrates well how when the queues are highly synchronised, it is better to remain synchronized rather than deviate, even if the synchronisation is suboptimal in terms of stability. The complete proof is provided in Appendix C.

Queues following this strategy accumulate packets during the first stage, and clear more packets than they receive during the second stage. The value of $\alpha$ is tuned so that the queues still accumulate a linear portion of packets during each time window. For those appropriate $\alpha$, the system is unstable.

Now suppose that queue $i$ deviates from the strategy and plays a fixed action $p \in \mathcal{P}\left([K]\right)$. In the first stage of each time window, queue $i$ can clear a bit more packets than it would by not deviating. However, during the second stage, it is no longer synchronised with the other queues and collides with them a large number of times. Because of those collisions, it will accumulate many packets. In the detailed analysis, we demonstrate that, in the end, for appropriate values of $\alpha$, queue $i$ accumulates more packets than it would have without deviating. $\square$

According to Theorem 4, the factor $\frac{e}{e-1}$ can be seen as the price of anarchy of the problem, as for slacks below, the worst Nash equilibria might be unstable. On the other hand, it is known that for any slack above 1, there exists a centralized stable strategy. This centralized strategy actually consists in queues playing the same joint probability at each time step, independently from the number of accumulated packets. As a consequence, it is also a correlated equilibrium of the patient game and 1 can be seen as the correlated price of stability.

## 4 A decentralized algorithm

This section describes the decentralized algorithm ADEQUA, whose pseudocode is given in Algorithm 1. Due to space constraints, all the proofs are postponed to Appendix D. ADEQUA assumes all queues *a priori* know the number $N$ of queues in the game and have a unique rank or *id* in $[N]$. Moreover, the existence of a shared randomness between all queues is assumed. The *id* assumption is required to break the symmetry between queues and is classical in multiplayer bandits without collision information. On the other side, the shared randomness assumption is equivalent to the knowledge of a common seed for all queues, which then use this common seed for their random generators. A similar assumption is used in multiplayer bandits [Bubeck et al., 2020].

ADEQUA is inspired by the celebrated $\varepsilon$-greedy strategy. With probability $\varepsilon_t = (N + K)t^{-\frac{1}{5}}$, at each time step, queues explore the different parameters $\lambda_i$ and $\mu_i$ as described below. Otherwise with probability $1 - \varepsilon_t$, they exploit the servers. Each queue $i$ then sends a packet to a server following a policy solely computed from its local estimates $\hat{\lambda}^i, \hat{\mu}^i$ of the problem parameters $\lambda$ and $\mu$. The shared randomness is here used so that exploration simultaneously happens for all queues. If exploration/exploitation was not synchronized between the queues, an exploiting queue could collide with an exploring queue, biasing the estimates $\hat{\lambda}^i, \hat{\mu}^i$ of the latter.

---
**Algorithm 1:** ADEQUA
---
**input :** $N$ (number of queues), $i \in [N]$ (queue *id*)

1 **for** $t = 1, \ldots, \infty$ **do**
2     $\hat{P} \leftarrow \phi(\hat{\lambda}, \hat{\mu})$ and $\hat{A} \leftarrow \psi(\hat{P})$ where $\phi$ and $\psi$ are resp. defined by Equations (3) and (4)
3     Draw $\omega_1 \sim \text{Bernoulli}((N+K)t^{-\frac{1}{5}})$ and $\omega_2 \sim \mathcal{U}(0,1)$         `// shared randomness`
4     **if** $\omega_1 = 1$ **then** EXPLORE$(i)$                     `// exploration`
5     **else** Pull $\hat{A}(\omega_2)(i)$                        `// exploitation`
6 **end**
---

**Exploration.** When exploring, queues choose either to explore the servers' parameters $\mu_k$ or the other queues' parameters $\lambda_i$ as described in Algorithm 2 below. In the former case, all queues choose different servers at random (if they have packets to send). These rounds are used to estimate the servers means: $\hat{\mu}_k^i$ is the empirical mean of server $k$ observed by the queue $i$ for such rounds. Thanks to the shared randomness, queues pull different servers here, making the estimates unbiased.

In the latter case, queues explore each other in a pairwise fashion. When queues $i$ and $j$ explore each other at round $t$, each of them sends their **most recent** packet to some server $k$, chosen uniformly at random, if and only if a packet appeared during round $t$. In that case, we say that *the queue $i$ explores $\lambda_j$ (and vice versa)*. To make sure that $i$ and $j$ are the only queues choosing the server $k$ during this step, we proceed as follows:

- queues sample a matching $\pi$ between queues at random. To do so, the queues use the same method to plan an all-meet-all (or round robin) tournament, for instance Berger tables [Berger, 1899], and choose uniformly at random which round of the tournament to play. If the number of queues $N$ is odd, in each round of the tournament, one queue remains alone and does nothing.

- the queues draw the same number $l \sim \mathcal{U}([K])$ with their shared randomness. For each pair of queues $(i, j)$ matched in $\pi$, associate $k_{(i,j)} = l + \min(i,j) \pmod{K} + 1$ to this pair. The queues $i$ and $j$ then send to the server $k_{(i,j)}$.

As we assumed that the server breaks ties in the packets' age uniformly at random, the queue $i$ clears with probability $(1 - \frac{\lambda_j}{2})\bar{\mu}$, where $\bar{\mu} = \frac{1}{K}\sum_{k=1}^K \mu_k$. Thanks to this, $\lambda_j$ is estimated by queue $i$ as:

$$\hat{\lambda}_j^i = 2 - 2\hat{S}_j^i/\tilde{\mu}^i, \tag{2}$$

where $\tilde{\mu}^i = \frac{\sum_{k=1}^K N_k^i \hat{\mu}_k^i}{\sum_{k=1}^K N_k^i}$, $N_k^i$ is the number of *exploration* pulls of server $k$ by queue $i$ and $\hat{S}_j^i$ is the empirical probability of clearing a packet observed by queue $i$ when exploring $\lambda_j$.

---
**Algorithm 2:** EXPLORE
---
**input :** $i \in [N]$                                           `// queue id`

1 $k \leftarrow 0$
2 Draw $n \sim \mathcal{U}([N+K])$                        `// shared randomness`
3 **if** $n \leq K$ **then**                                `// explore` $\mu$
4     $k \leftarrow n + i \pmod{K} + 1$
5     Pull $k$ ;    Update $N_k$ and $\hat{\mu}_k$
6 **else**                                       `// explore` $\lambda$
7     Draw $r \sim \mathcal{U}([N])$ and $l \sim \mathcal{U}([K])$         `// shared randomness`
8     $j \leftarrow r^{\text{th}}$ opponent in the all-meet-all tournament planned according to Berger tables
9     $k \leftarrow l + \min(i,j) \pmod{K} + 1$
10     **if** $k \neq 0$ *and packet appeared at current time step* **then**     `// explore` $\lambda_j$ `on server` $k$
11        Pull $k$ with most recent packet ;    Update $\hat{S}_j$ and $\hat{\lambda}_j$ according to Equation (2)
12     **end**
13 **end**
---

**Remark 1.** *The packet manipulation when exploring $\lambda_j$ strongly relies on the servers tie breaking rules (uniformly at random). If this rule was unknown or not explicit, the algorithm can be adapted: when queue $i$ explores $\lambda_j$, queue $j$ instead sends the packet generated at time $t - 1$ (if it exists), while queue $i$ still sends the packet generated at time $t$. In that case, the clearing probability for queue $i$ is exactly $(1 - \lambda_j)\bar{\mu}$, allowing to estimate $\lambda_j$. Anticipating the nature of the round $t$ (exploration vs.*

*exploitation) can be done by drawing $\omega_1 \sim \text{Bernoulli}(\varepsilon_t)$ at time $t - 1$. If $\omega_1 = 1$, the round $t$ is exploratory and the packet generated at time $t - 1$ is then kept apart by the queue $j$.*

To describe the exploitation phase, we need a few more notations. We denote by $\mathfrak{B}_K$ the set of doubly stochastic matrices (non-negative matrices such that each of its rows and columns sums to 1) and by $\mathfrak{S}_K$ the set of permutation matrices in $[K]$ (a permutation matrix will be identified with its associated permutation for the sake of cumbersomeness).

A **dominant mapping** is a function $\phi : \mathbb{R}^N \times \mathbb{R}^K \rightarrow \mathfrak{B}_K$ which, from $(\lambda, \mu)$, returns a doubly stochastic matrix $P$ such that $\lambda_i < (P\mu)_i$ for any $i \in [N]$ if it exists (and the identity matrix otherwise).

A **BvN** (Birkhoff von Neumann) **decomposition** is a function $\psi : \mathfrak{B}_K \rightarrow \mathcal{P}(\mathfrak{S}_K)$ that associates to any doubly stochastic matrix $P$ a random variable $\psi(P)$ such that $\mathbb{E}[\psi(P)] = P$; stated otherwise, it expresses $P$ as a convex combination of permutation matrices. For convenience, we will represent this random variable as a function from $[0, 1]$ (equipped with the uniform distribution) to $\mathfrak{S}_K$.

Informally speaking, those functions describe the strategies queues would follow in the centralized case: a dominant mapping gives adequate marginals ensuring stability (since the queue $i$ clears in expectation $(P\mu)_i$ packets at each step, which is larger than $\lambda_i$ by definition), while a BvN decomposition describes the associated coupling to avoid collisions. Explicitly, the joint strategy is for each queue to draw a shared random variable $\omega_2 \sim \mathcal{U}(0, 1)$ and to choose servers according to the permutation $\psi(\phi(\lambda, \mu))(\omega_2)$

**Exploitation.** In a decentralized system, each queue $i$ computes a mapping $\hat{A}^i := \psi(\phi(\hat{\lambda}^i, \hat{\mu}^i))$ solely based on its own estimates $\hat{\lambda}^i, \hat{\mu}^i$. A shared variable $\omega_2 \in [0, 1]$ is then generated uniformly at random and queue $i$ sends a packet to the server $\hat{A}^i(\omega_2)(i)$. If all queues knew exactly the parameters $\lambda, \mu$, the computed strategies $\hat{A}^i$ would be identical and they would follow the centralized policy described above.

However, the estimates $(\hat{\lambda}^i, \hat{\mu}^i)$ are different between queues. The usual dominant mappings and BvN decompositions in the literature are non-continuous. Using those, even queues with close estimates could have totally different $\hat{A}^i$, and thus collide a large number of times, which would impede the stability of the system. Regular enough dominant mappings and BvN decompositions are required, to avoid this phenomenon. The design of $\phi$ and $\psi$ is thus crucial and appropriate choices are given in the following Sections 4.1 and 4.2. Nonetheless, they can be used in some black-box fashion, so we provide for the sake of completeness sufficient conditions for stability, as well as a general result depending on the properties of $\phi$ and $\psi$, in Appendix A.

**Remark 2.** *The exploration probability $t^{-\frac{1}{5}}$ gives the smallest theoretical dependency in $\Delta$ in our bound. A trade-off between the proportion of exploration rounds and the speed of learning indeed appears in the proof of Theorem 1. Exploration rounds have to represent a small proportion of the rounds, as the queues accumulate packets when exploring. On the other hand, if queues explore more often, the regime where their number of packets decreases starts earlier. A general stability result depending on the choice of this probability is given by Theorem 6 in Appendix A.*
*Yet in Section 5, taking a probability $t^{-\frac{1}{4}}$ empirically performs better as it speeds up the exploration.*

## 4.1 Choice of a dominant mapping

Recall that a dominant mapping takes as inputs $(\lambda, \mu)$ and returns, if possible, a doubly stochastic matrix $P$ such that

$$\lambda_i < \textstyle\sum_{k=1}^K P_{i,k}\mu_k \text{ for all } i \in [N].$$

The usual dominant mappings sort the vector $\lambda$ and $\mu$ in descending orders [Marshall et al., 1979]. Because of this operation, they are non-continuous and we thus need to design a regular dominant mapping satisfying the above property. Inspired by the $\log$-barrier method, it is done by taking the minimizer of a strongly convex program as follows

$$\phi(\lambda, \mu) = \underset{P \in \mathfrak{B}_K}{\arg\min} \max_{i \in [N]} -\ln\Big(\sum_{j=1}^K P_{i,j}\mu_j - \lambda_i\Big) + \frac{1}{2K}\|P\|_2^2. \tag{3}$$

Although the objective function is non-smooth because of the max operator, it enforces fairness between queues and leads to a better regularity of the $\arg\min$.

**Remark 3.** *Computing $\phi$ requires solving a non-smooth strongly convex minimization problem. This cannot be computed exactly, but a good approximation can be quickly obtained using the scheme described in Appendix B. If this approximation error is small enough, it has no impact on the stability bound of Theorem 5. It is thus ignored for simplicity, i.e., we assume in the following that $\phi(\lambda, \mu)$ is exactly computed at each step.*

As required, $\phi$ always returns a matrix $P$ satisfying that $\lambda < P\mu$ if possible, since otherwise the objective is infinite (and in that case we assume that $\phi$ returns the identity matrix). Moreover, the objective function is $\frac{1}{K}$-strongly convex, which guarantees some regularity of the $\arg\min$, namely local-Lipschitzness, leading to Lemma 1 below.

**Lemma 1.** *For any $(\lambda, \mu)$ with positive margin $\Delta$ (defined in Equation (1)), if $\|(\hat{\lambda} - \lambda, \hat{\mu} - \mu)\|_\infty \leq c_1\Delta$, for any $c_1 < \frac{1}{2\sqrt{e}+2}$, then*

$$\|\phi(\hat{\lambda}, \hat{\mu}) - \phi(\lambda, \mu)\|_2 \leq \frac{c_2 K}{\Delta}\|(\hat{\lambda} - \lambda, \hat{\mu} - \mu)\|_\infty,$$

*where $c_2 = \frac{4}{(1-2c_1)/\sqrt{e}-2c_1}$. Moreover, denoting $\hat{P} = \phi(\hat{\lambda}, \hat{\mu})$, it holds for any $i \in [N]$,*

$$\lambda_i \leq \textstyle\sum_{k=1}^{K} \hat{P}_{i,k}\mu_k - \left(\frac{1-2c_1}{\sqrt{e}} - 2c_1\right)\Delta.$$

The first property guarantees that if the queues have close estimates, they also have close doubly stochastic matrices $\hat{P}$. Moreover, the second property guarantees that any queue should clear its packets with a margin of order $\Delta$, in absence of collisions.

**Remark 4.** *An alternative dominant mapping without the regularizing term in Equation (3) can also be proposed. Yet, its local Lipschitz bound would also depend on the smallest difference between the $\lambda_i$ or the $\mu_i$, which can be arbitrarily small. If two parameters $\lambda_i$ or $\mu_i$ are equal, this choice of dominant mapping might lead to unstable policies. Using a regularization term in Equation (3) thus avoids this problem, although a smaller dependency might be possible without this regularization term when the parameters $\lambda_i$ and $\mu_i$ are very distinct.*

### 4.2 Choice of a Birkhoff von Neumann decomposition

Given a doubly stochastic matrix $\hat{P}$, Birkhoff algorithm returns a convex combination of permutation matrices $P[j]$ such that $\hat{P} = \sum_j z[j]P[j]$. The classical version of Birkhoff algorithm is non-continuous in its inputs and it even holds for its extensions as the one proposed by Dufossé et al. [2018]. Yet it can be smartly modified as in ORDERED BIRKHOFF, described in Algorithm 3, to get a regular BvN decomposition defined as follows for any $\omega \in (0, 1)$:

$$\psi(P)(\omega) = P[j_\omega] \tag{4}$$

where $P = \sum_j z[j]P[j]$ is the decomposition returned by ORDERED BIRKHOFF algorithm

$$\text{and } j_\omega \text{ verifies } \sum_{j \leq j_\omega} z[j] \leq \omega < \sum_{j \leq j_\omega+1} z[j].$$

For a matrix $P$ in the following, its support is defined as $\operatorname{supp}(P) = \{(i, j) \mid P_{i,j} \neq 0\}$.

---

**Algorithm 3:** ORDERED BIRKHOFF

**input :** $\hat{P} \in \mathfrak{B}_K$ (doubly stochastic matrix), $C \in \mathbb{R}^{K \times K}$ (cost matrix)

1   $j \leftarrow 1$
2   **while** $\hat{P} \neq \mathbf{0}$ **do**
3     $C_{i,k} \leftarrow +\infty$ for all $(i, k) \notin \operatorname{supp}(\hat{P})$        `// remove edge (i, k) in induced graph`
4     $P[j] \leftarrow \text{HUNGARIAN}(C)$        `// matching with minimal cost w.r.t. C`
5     $z[j] \leftarrow \min_{(i,k)\in\operatorname{supp}(P[j])} \hat{P}_{i,k}$
6     $\hat{P} \leftarrow \hat{P} - z[j]P[j]$    and    $j \leftarrow j + 1$
7   **end**
8   **return** $(z[j], P[j])_j$

---

Obviously $\mathbb{E}_{\omega \sim \mathcal{U}(0,1)}[\psi(P)(\omega)] = P$ and permutations avoid collisions between queues. The difference with the usual Birkhoff algorithm happens at Line 4. Birkhoff algorithm usually computes any perfect matching in the graph induced by the support of $\hat{P}$ at the current iteration. This is often done with the Hopcroft-Karp algorithm, while it is here done with the Hungarian algorithm with respect to some cost matrix $C$. Although using the Hungarian algorithm slightly increases the computational complexity of this step ($K^3$ instead of $K^{2.5}$), it ensures to output the permutation matrices $P[j]$ according to a fixed order defined below.

**Definition 1.** *A cost matrix $C$ induces an order $\prec_C$ on the permutation matrices defined, for any $P, P' \in \mathfrak{S}_K$ by*

$$P \prec_C P' \quad \textit{iff} \quad \sum_{i,j} C_{i,j} P_{i,j} < \sum_{i,j} C_{i,j} P'_{i,j}.$$

This order might be non-total as different permutations can have the same cost. However, if $C$ is drawn at random according to some continuous distribution, this order is total with probability 1. The order $\prec_C$ has to be the same for all queues and is thus determined beforehand for all queues.

**Lemma 2.** *Given matrices $C \in \mathbb{R}^{K \times K}$ and $P \in \mathfrak{B}_K$, ORDERED BIRKHOFF outputs a sequence $(z[j], P[j])_j$ of length at most $K^2 - K + 1$, such that*

$$P = \sum_j z[j]P[j], \textit{ where for all } j, \ z[j] > 0 \textit{ and } P[j] \in \mathfrak{S}_K.$$

*Moreover if the induced order $\prec_C$ is total, $z[j]$ is the $j$-th non-zero element of the sequence $(z_l(P))_{1 \le l \le K!}$ defined by*

$$z_j(P) = \min_{(i,k) \in \text{supp}(P_j)} \left( P - \sum_{l=1}^{j-1} z_l(P)P_l \right)_{i,k} \tag{5}$$

*where $(P_j)_{1 \le j \le K!}$ is a $\prec_C$-increasing sequence of permutation matrices, i.e., $P_j \prec_C P_{j+1}$ for all $j$.*

Lemma 2 is crucial to guarantee the regularity of $\psi$, given by Lemma 3.

**Lemma 3.** *Consider $\psi$ defined as in Equation (4) with a cost matrix $C$ inducing a total order $\prec_C$, then for any doubly stochastic matrices $P, P'$*

$$\int_0^1 \mathbb{1}\left(\psi(P)(\omega) \neq \psi(P')(\omega)\right) d\omega \le 2^{2(K^2 - K + 1)} \|P - P'\|_\infty.$$

Lemma 3 indeed ensures that the probability of collision between two queues remains small when they have close estimates. Unfortunately, the regularity constant is exponential in $K^2$, which yields a similar dependency in the stability bound of Theorem 5. The existence of a BvN decomposition with polynomial regularity constants remains unknown, even without computational considerations. The design of a better BvN decomposition is left open for future work and would directly improve the stability bounds, using the general result given by Theorem 6 in Appendix A. The number of accumulated packets yet remain reasonably small in the experiments of Section 5, suggesting that the bound given by Lemma 3 is not tight and might be improved in future work.

### 4.3 Stability guarantees

This section finally provides theoretical guarantees on the stability of the system when all queues follow ADEQUA. The success of ADEQUA relies on the accurate estimation of all problem parameters by the queues, given by Lemma 9 in Appendix D.4. After some time $\tau$, the queues have tight estimations of the problem parameters. Afterwards, they clear their packets with a margin of order $\Delta$, thanks to Lemmas 1 and 3. This finally ensures the stability of the system, as given by Theorem 5.

**Theorem 5.** *For any $\eta > 1$, the system where all queues follow ADEQUA, for any queue $i$ and any $r \in \mathbb{N}$, there exists a constant $C_r$ depending only on $r$ such that*

$$\mathbb{E}[(Q_t^i)^r] \le C_r K N \left( \frac{N^{\frac{5}{2}} K^{\frac{5}{2}} 2^{5(K^2 - K + 1)}}{(\min(1, K\bar{\mu})\underline{\lambda})^{\frac{5}{4}} \Delta^5} \right)^r, \quad \textit{for all } t \in \mathbb{N}.$$

*As a consequence, for any $\eta > 1$, this decentralized system is strongly stable.*

Despite yielding an exponential dependency in $K^2$, this anytime bound leads to a first decentralized stability result when $\eta \in (1, \frac{e}{e-1})$, which closes the stability gap left by previous works. Moreover it can be seen in the proof that the asymptotic number of packets is much smaller. It actually converges, in expectation, to the number of packets the queues would accumulate if they were following a stable centralized strategy from the beginning. As already noted by Krishnasamy et al. [2016] for a single queue, the number of packets first increases during the learning phase and then decreases once the queues have tight enough estimations, until reaching the same state as in the perfect knowledge centralized case. This is empirically confirmed in Section 5.

## 5 Simulations

Figures 2 and 3 compare on toy examples the stability of queues, when either each of them follows the no-regret strategy EXP3.P.1, or each queue follows ADEQUA. For practical considerations, we choose the exploration probability $\varepsilon_t = (N + K)t^{-\frac{1}{4}}$ for ADEQUA, as the exploration is too slow with $\varepsilon_t$ of order $t^{-\frac{1}{5}}$.

These figures illustrate the evolution of the average queue length on two different instances with $N = K = 4$. The code for the experiments is available at `gitlab.com/f_sen/queuing_systems`.

In the first instance shown in Figure 2, for all $i \in [N]$, $\lambda_i = (N + 1)/N^2$. Moreover $\mu_1 = 1$ and for all $i \geq 2$, $\mu_i = (N - 1)/N^2$. Here $\eta < 2$ and no-regret strategies are known to be unstable [Gaitonde and Tardos, 2020a]. It is empirically confirmed as the number of packets in each queue diverges when they follow EXP3.P.1. Conversely, when the queues follow ADEQUA, after a learning phase, the queues reach equilibrium and all succeed in clearing their packets.

In the second instance shown in Figure 3, for all $i \in [N]$, $\lambda_i = 0.55 - 0.1 \cdot i$ and $\mu_i = 2.1\lambda_i$. Here $\eta > 2$ and both strategies are known to be stable, which is again empirically confirmed. However, ADEQUA requires more time to learn the different parameters, suggesting that individual no-regret strategies might be better on easy instances where $\eta > 2$.

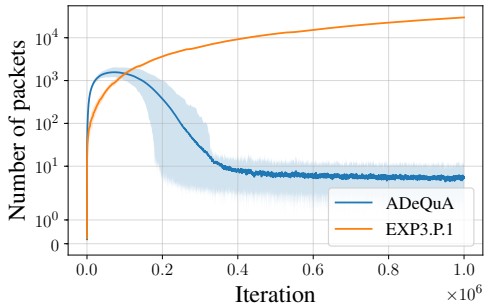
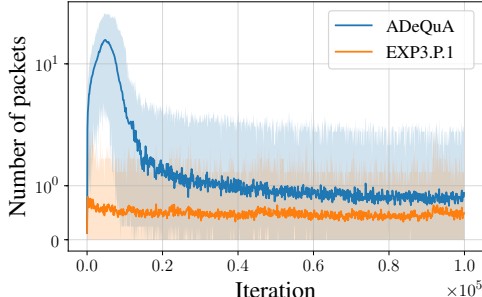

Figure 2: Hard instance, $\eta < 2$.

Figure 3: Easy instance, $\eta > 2$.

## 6 Conclusion

In this work, we showed that minimizing a more patient version of regret was not necessarily stable when the system's slack is smaller than two and we argued that some level of cooperation was then required between learning queues to reach stability. We presented the first decentralized learning algorithm guaranteeing stability of any queuing system with a slack larger than 1. Our stability bound presents an exponential dependency in the number of queues and remains open for improvement, e.g., through a better dominant mapping/BvN decomposition or a tighter analysis of ours. The proposed algorithm relies heavily on synchronisation between the queues, which all start the game simultaneously and share a common time discretisation. In particular, the shared randomness assumption merely results from this synchronisation when the players use a common random seed. Stability of asynchronous queues thus remains open for future work, for which Glauber dynamics approaches used in scheduling problems might be of interest [see e.g., Shah and Shin, 2012].

## Acknowledgements

F. Sentenac was supported by IP PARIS' PhD Funding. E. Boursier was supported by an AMX scholarship. V. Perchet acknowledges support from the French National Research Agency (ANR) under grant number #ANR-19-CE23-0026 as well as the support grant, as well as from the grant "Investissements d'Avenir" (LabEx Ecodec/ANR-11-LABX-0047).

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
