# A    General version of Theorem 5

ADEQUA is described for specific choices of the functions $\phi$ and $\psi$ given by Sections 4.1 and 4.2. It yet uses them in a black box fashion and different functions can be used, as long as they verify some key properties. This section provides a general version of Theorem 5, when the used dominant mapping and BvN decomposition respect the properties given by Assumptions 1 and 2.

**Assumption 1** (regular dominant mapping). *There are constants $c_1, c_2 > 0$ and a norm $\|\cdot\|$ on $\mathbb{R}^{K \times K}$, such that if $\|(\hat{\lambda} - \lambda, \hat{\mu} - \mu)\|_\infty \leq c_1 \Delta$, then*

$$\|\phi(\hat{\lambda}, \hat{\mu}) - \phi(\lambda, \mu)\| \leq L_\phi \cdot \|(\hat{\lambda} - \lambda, \hat{\mu} - \mu)\|_\infty.$$

*Moreover, $\hat{P} = \phi(\hat{\lambda}, \hat{\mu})$ is doubly stochastic and for any $i \in [N]$,*

$$\lambda_i \leq \sum_{k=1}^{K} \hat{P}_{i,k} \mu_k - c_2 \Delta.$$

**Assumption 2** (regular BvN decomposition). *Consider the same norm $\|\cdot\|$ as Assumption 1 on $\mathbb{R}^{K \times K}$. For any doubly stochastic matrices $P, P'$*

$$\int_0^1 \psi(P)(\omega) \mathrm{d}\omega = P$$

$$\text{and } \int_0^1 \mathbb{1}\left(\psi(P)(\omega) \neq \psi(P')(\omega)\right) \mathrm{d}\omega \leq L_\psi \cdot \|P - P'\|.$$

Lemmas 1 and 3 show that the functions described in Sections 4.1 and 4.2 verify Assumptions 1 and 2 with the constants $L_\phi$ and $L_\psi$ respectively of order $\frac{K}{\Delta}$ and $2^{2K^2}$ with the norm $\|\cdot\|_\infty$. Designing a dominant mapping and a BvN decomposition with smaller constants $L_\phi$ and $L_\psi$ is left open for future work. It would lead to a direct improvement of the stability bound, as shown by Theorem 6.

**Theorem 6.** *Assume all queues follow ADEQUA, using an exploration probability $\varepsilon_t = xt^{-\alpha}$ with $x > 0, \alpha \in (0,1)$ and functions $\phi$ and $\psi$ verifying Assumptions 1 and 2 with the constants $L_\phi, L_\psi$. The system is then strongly stable and for any $r \in \mathbb{N}$, there exists a constant $C_r$ such that:*

$$\mathbb{E}[(Q_t^i)^r] \leq C_r \left( \frac{x^{r/\alpha}}{\Delta^{r/\alpha}} + KN \left( \frac{N^2 K L_\phi^2 L_\psi^2}{\min(1, K\bar{\mu})\underline{\lambda}\Delta^2 x} \right)^{\frac{r}{1-\alpha}} \right), \quad \text{for all } t \in \mathbb{N}$$

The proof directly follows the lines of the proof of Theorem 5 in Appendix D.4 and is thus omitted here. From this version, it can be directly deduced that $\alpha = \frac{1}{5}$ gives the best dependency in $\Delta$ for ADEQUA. Moreover the best choice for $x$ varies with $r$. When $r \to \infty$, it actually is $x = N^{\frac{2}{5}} K^{\frac{3}{5}} 2^{\frac{4}{5}K^2}$ for ADEQUA. The choice $x = N + K$ is preferred for simplicity and still yields quite similar problem dependent bounds.

# B    Efficient computation of $\phi$

As mentioned in Section 4.1, computing exactly $\phi(\hat{\lambda}, \hat{\mu})$ is not possible. Even efficiently approximating it is not obvious, as the function to minimize is neither smooth nor Lipschitz. We here describe how an approximation of $\phi$ can be efficiently computed with guarantees on the approximation error.

First define the empirical estimate of the margin $\Delta$:

$$\hat{\Delta} := \min_{k \in [N]} \frac{1}{k} \left( \sum_{i=1}^{k} \hat{\mu}_{(i)} - \hat{\lambda}_{(i)} \right).$$

It can be computed in time $\mathcal{O}\left(N \log(N)\right)$ as it only requires to sort the vectors $\hat{\lambda}$ and $\hat{\mu}$. If $\hat{\Delta} \leq 0$, then the value of the optimization problem is $+\infty$ and any matrix can be returned. Assume in the following $\hat{\Delta} > 0$. Similarly to the proof of Lemma 1, it can be shown that the value of the optimization problem is smaller than $-\ln(\hat{\Delta}/\sqrt{e})$. Noting by $\mathfrak{B}_K$ the set of $K \times K$ doubly stochastic matrices, the optimization problem given by Equation (3) is then equivalent to

$$\underset{P \in \mathcal{X}}{\arg\min} \, g(P), \tag{6}$$

where

$$\mathcal{X} = \left\{ P \in \mathfrak{B}_K \mid \forall i \in [N], \sum_{j=1}^{K} P_{i,j}\mu_j - \lambda_i \geq \frac{\hat{\Delta}}{\sqrt{e}} \right\},$$

$$\text{and } g(P) = \max_{i \in [N]} -\ln(\sum_{j=1}^{K} P_{i,j}\mu_j - \lambda_i) + \frac{1}{2K}\|P\|_2^2.$$

Thanks to this new constraint set, the objective function of Equation (6) is now $(\frac{\sqrt{e}}{\Delta} + 1)$-Lipschitz. We can now use classical results for Lipschitz strongly convex minimization to obtain convergence rates of order $\frac{1}{t}$ for the projected gradient descent algorithm [see e.g., Bubeck, 2014, Theorem 3.9]. These results yet assume that the projection on the constraint set can be exactly computed in a short time. This is not the case here, but it yet can be efficiently approximated using interior point methods [see e.g., Bubeck, 2014, Section 5.3], which has a linear convergence rate. If this approximation is good enough, similar convergence guarantees than with exact projection can be shown similarly to the original proof.

Algorithm 4 then describes how to quickly estimate $\phi(\hat{\lambda}, \hat{\mu})$, where $\hat{\Pi}_{\mathcal{X}}$ returns an approximation of the orthogonal projection on the set $\mathcal{X}$ and $\partial g$ is a sub-gradient of $g$. It uses an averaged value of the different iterates, as the last iterate does not have good convergence guarantees.

---

**Algorithm 4:** Compute $\phi$

**input :** function $g$, constraint set $\mathcal{X}$, $P^0 \in \mathcal{X}$

1   $P, \hat{P} \leftarrow P^0$

2   **for** $t = 1, \ldots, n$ **do**

3      $P \leftarrow \hat{\Pi}_{\mathcal{X}}\left(P - \frac{2N}{(t+1)}\partial g(P)\right)$                          `// approximated projection`

4      $\hat{P} \leftarrow \frac{t}{t+2}\hat{P} + \frac{2}{t+2}P$

5   **end**

6   **return** $\hat{P}$

---

In practice, the approximation can even be computed faster by initializing $P^0$ in Algorithm 4 with the solution of the previous round $t - 1$.

## C   Unstable No-Policy regret system example

---

**Algorithm 5:** Unstable No-policy regret system example

**input :** $w_k$, $N$, $\alpha$, $\lambda = (1/N, \ldots, 1/N)$, $\mu = (2(N-d)/N^2, \ldots, 2(N-d)/N^2)$

1   **for** $k = 1, \ldots, \infty$ **do**

2      **for** $t = 1, \ldots, \lceil \alpha w_k \rceil$ **do**

3          Queues $2i$ and $2i + 1$ play server $2i + t \pmod{N}$                `// stage 1`

4      **end**

5      **for** $t = \lceil \alpha w_k \rceil + 1, \ldots, w_k$ **do**

6          Queue $i$ plays server $i + t \pmod{N}$                           `// stage 2`

7      **end**

8   **end**

---

**Lemma 4.** *Consider the system where the queues play according to the policy described in Algorithm 5 over successive windows of length $w_k = k^2$. If $\alpha > 1 - \frac{d}{N-d}$, the system is not stable.*

*Proof.* Note that the system is equivalent to a system where each queue or pair of queue would always pick the same server. For simplicity, the analysis deals with that equivalent system. Also, wlog, we analyse the subsystem with the two first queues and the two first servers. Let $\{B_t^i\}_{i \in [n], t \geq 1}$ be the independent random variables indicating the arrival of a packet on queue $i$ at time $t$, $\{S_t^i\}_{i \in [n], t \geq 1}$ be the indicators that server $j$ would clear a packet at iteration $\ell$ if one were sent to it. For each queue $i \in [N]$ and $t \geq 0$, we have by Chernoff bound

$$\Pr\left(\left|\sum_{t=1}^{\ell} B_t^i - \lambda_i \ell\right| \geq \sqrt{\ell \ln(\ell)}\right) \leq \frac{2}{\ell^2}.$$

The same holds for each queue, thus the probability that this event happens for queue 1 or queue 2 is at most, $\frac{4}{\ell^2}$. As it is summable in $\ell$, The Borel-Cantelli Lemma implies that, for large enough $\ell$, almost surely, for any $i \in [2]$:

$$\sum_{\ell=1}^{\ell} B_t^i = \lambda_i \ell \pm \tilde{\mathcal{O}}\left(\sqrt{\ell}\right). \tag{7}$$

Let $W_k = \sum_{i=1}^k w_i$. Note that $W_k = \Theta\left(k^3\right) = \Theta\left(w_k^{3/2}\right)$. Again by Chernoff bound and Borel-Cantelli, for large enough $k$, almost surely, for any $i \in \{1, 2\}$:

$$\sum_{t=W_{k-1}}^{W_{k-1}+\lceil \alpha w_k \rceil} S_t^i = \mu_i \alpha w_k \pm \tilde{\mathcal{O}}\left(\sqrt{w_k}\right), \qquad \sum_{t=W_{k-1}+\lceil \alpha w_k \rceil}^{W_k} S_t^i = \mu_i(1-\alpha)w_k \pm \tilde{\mathcal{O}}\left(\sqrt{w_k}\right). \tag{8}$$

Thus, for any large enough $k$, the total number of packet in both queues at time $W_k$ is almost surely lower bounded as:

$$Q_{W_k}^1 + Q_{W_k}^2 \geq \sum_{t=1}^{W_k}(B_t^1 + B_t^2) - \sum_{t=1}^{W_k} S_t^1 - \sum_{l=1}^{k}\left(\sum_{t=W_{l-1}+\lceil \alpha w_l \rceil}^{W_l} S_t^2\right) \tag{9}$$

$$\geq \left[\frac{2}{N} - \frac{2(N-d)}{N^2} - (1-\alpha)\frac{2(N-d)}{N^2}\right]W_k - \tilde{\mathcal{O}}\left(W_k^{2/3}\right) \tag{10}$$

$$\geq \frac{2\left[\alpha(N-d) - (N-2d)\right]}{N^2}W_k - \tilde{\mathcal{O}}\left(W_k^{2/3}\right) \tag{11}$$

which is a diverging function of $W_k$. Note that this result also holds for any two pair of queues $(2i-1, 2i)$, with $i \in [N/2]$.

$\square$

**Lemma 5.** *Consider the same setting as in Lemma 4. For any $i \in [N]$, for any large enough $k$, queue $i$ clears*

$$\left(\frac{N-d}{N^2} + (1-\alpha)\frac{N-d}{N^2} + o(1)\right)w_k$$

*packets almost surely over window $w_k$.*

*Proof.* The proof starts by showing that for any large enough $t$, all the queues hold roughly the same number of packets. Then, as they receive roughly the same number of packets over a time window and we can compute the approximate total number of packets cleared, the results follows.

Let $T_i^t$ be the age of the oldest packet in queue $i$ at time $t$. By Chernoff bound,

$$\mathrm{P}(|T_i^t - NQ_i^t| \geq N\sqrt{t\ln(t)}) \leq \frac{2}{t^2}.$$

Thus, using the Borel-Cantelli lemma, for any queue $i$, almost surely, for any large enough $k$ and any $t \in [W_{k-1}+1, W_k]$,

$$|T_i^t - NQ_i^t| \leq N\sqrt{t\ln(t)} = \tilde{\mathcal{O}}(w_k^{3/4}). \tag{12}$$

For any $(i, j) \in [N]^2$, define

$$\phi_t^+(i,j) := \left(Q_t^i - Q_t^j - 2N\sqrt{t\ln(t)}\right)_+ \text{ and } \phi_t^-(i,j) := \left(Q_t^i - Q_t^j + 2N\sqrt{t\ln(t)}\right)_-.$$

Let $C_t^i$ be the indicator function that queue $i$ clears a packet at iteration $t$. Note that for any large enough $t$, $\phi_t^+(i,j)$ is a supermartingale. Indeed,

$$\mathbb{E}[\phi_{t+1}^+(i,j)|\phi_{1:t}^+(i,j)] \leq \phi_t^+(i,j) + \mathbb{E}[B_t^i - B_t^j|\phi_{1:t}^+(i,j)] - \mathbb{E}[C_t^i - C_t^j|\phi_{1:t}^+(i,j)]$$
$$\leq \phi_t^+(i,j).$$

The second inequality comes from Equation (12), that implies that for any large enough $t$, if $\phi_t^+(i,j)$ is strictly positive, queue $i$ holds the oldest packet and thus clears one with higher probability than queue $j$. By the same arguments, $\phi_t^-(i,j)$ a submartingale. Also, $|\phi_{t+1}^+(i,j) - \phi_t^+(i,j)| \leq 2(N+1)$ for any $t \geq 0$, and the same holds for $\phi_t^-(i,j)$. Let $\tau_{ij}$ be the stopping time of the smallest iteration after which Equation (12) always holds for queues $i$ and $j$. By Azuma-Hoeffding's inequality,

$$\Pr\left(\phi_\ell^+(i,j) - \phi_{\tau_{ij}}^+(i,j) \geq 3(N+1)\sqrt{\ell\ln(\ell)}\right) \leq \frac{2}{\ell^2}$$

and

$$\Pr\left(\phi_\ell^-(i,j) - \phi_{\tau_{ij}}^+(i,j) \leq -3(N+1)\sqrt{\ell\ln(\ell)}\right) \leq \frac{2}{\ell^2}.$$

This, together with a union bound and Borel-Cantelli's Lemma implies that almost surely, for any large enough $t$, for any $(i,j) \in [N]^2$

$$Q_t^i - Q_t^j = \tilde{\mathcal{O}}\left(\sqrt{t}\right). \tag{13}$$

This with Equation (9) implies that for any large enough $k$, for any $i \in [N]$, almost surely,

$$Q_{W_k}^i \geq \frac{[\alpha(N-d) - (N-2d)]}{N^2} W_k - \tilde{\mathcal{O}}\left(W_k^{2/3}\right).$$

This means that for any large enough $k$, every queue holds at least one packet over the whole window $w_k$. This and Equation (8) is already enough to show that for any time-window $w_k$, for any large enough $k$, the total number of packets cleared by any couple of queue $(2i-1, 2i)$, $i \in [N/2]$ is:

$$2\left(\frac{N-d}{N^2} + (1-\alpha)\frac{N-d}{N^2}\right) w_k + \tilde{\mathcal{O}}\left(\sqrt{w_k}\right).$$

During time window $w_k$, according to Equation (7), both every queue receives $\alpha w_k/N + \tilde{\mathcal{O}}\left(w_k^{3/4}\right)$ packets almost surely for any large enough $k$. Equation (13) implies that for any $i \in [N/2]$

$$Q_{W_k}^{2i-1} - Q_{W_k}^{2i} = \tilde{\mathcal{O}}\left(w_k^{3/4}\right) \text{ and } Q_{W_{k-1}}^{2i-1} - Q_{W_{k-1}}^{2i} = \tilde{\mathcal{O}}\left(w_k^{3/4}\right).$$

Therefore, over each time-window $w_k$, for any large enough $k$, each queue clears

$$\left(\frac{N-d}{N^2} + (1-\alpha)\frac{N-d}{N^2} + o(1)\right) w_k$$

packets almost surely. $\square$

**Lemma 6.** *Consider again the system where the queues play according to the policy described in Algorithm 5 over successive windows of length $w_k = k^2$. If $\alpha < 1 - \frac{1}{N-1}$, the queues have $o\left(w_k\right)$ policy regret in all but finitely many of the windows.*

Wlog, let us consider that queue 1 deviates, and plays at every iteration a server chosen from the probability distribution $\mathbf{p} = (p_1, ..., p_N)$, with $p_i$ the probability to play server $i$. To upper bound the number of packets queue 1 clears over each time window, we can assume it always has priority over queue 2 and ignore it in the analysis.

Before proving Lemma 6, we prove the following technical one.

**Lemma 7.** *Consider that a queue deviates from the strategy considered in Lemma 6 and plays at every iteration a server chosen from the probability distribution $\mathbf{p} = (p_1, ..., p_N)$, with $p_i$ the probability to play server $i$. For any large enough $k$, almost surely, the number of packets the deviating queue clears of the first stage of the $k^{th}$ window is*

$$\left(\frac{1}{2} + \frac{1}{N}\right)\frac{2(N-d)}{N^2}\alpha w_k + \tilde{\mathcal{O}}\left(w_k^{3/4}\right).$$

*Proof.* The proof starts by showing that for any large enough $t$, every non-deviating queue holds approximately the same number of packets.

Fist note that for any large enough $t$, Equation (12) still holds surely for any queue $i$. For any $(i,j) \in \{3,\ldots,N\}^2$, define

$$\phi_\ell^+(i,j) := \left(Q_{\lceil \ell N\rceil}^i - Q_{\lceil \ell N\rceil}^j - 4N\sqrt{\lceil \ell N\rceil \ln(\lceil \ell N\rceil)}\right)_+$$

and

$$\phi_\ell^-(i,j) := \left(Q_{\lceil \ell N\rceil}^i - Q_{\lceil \ell N\rceil}^j + 4N\sqrt{\lceil \ell N\rceil \ln(\lceil \ell N\rceil)}\right)_-.$$

For any interval $[\lceil \ell N\rceil, \lceil (\ell+1)N\rceil]$ where Equation (12) holds for queues $1, i$ and $j$, if $\phi_\ell^+(i,j)$ is strictly positive, then

$$\mathbb{E}\left[\sum_{t=\lceil \ell N\rceil}^{\lceil (\ell+1)N\rceil} C_t^j - C_t^i \middle| \phi_{1:t}^+(i,j)\right] \leq 0.$$

Indeed, if $\phi_\ell^+(i,j)$ is strictly positive and Equation (12) holds, queue $i$ holds the oldest packets throughout the interval. Also, queue $i$ and queue $j$ collide with queue 1 the same number of times over the interval in expectation, and if at one iteration of the interval, queue 1 holds an older packet than queue $i$, it holds an older packet than queue $j$ over the whole interval. Thus $\phi_\ell^+(i,j)$ is a submartingale. By the same arguments, $\phi_\ell^+(i,j)$ is a supermartingale. Also, $|\phi_{\ell+1}^+(i,j) - \phi_\ell^+(i,j)| \leq 4(N+1)^2$ and the same holds for $\phi_\ell^-(i,j)$. Finishing with the same arguments used to prove Equation (13), almost surely, for any $(i,j) \in \{3,\ldots,N\}^2$,

$$Q_t^i - Q_t^j = \tilde{\mathcal{O}}\left(\sqrt{t}\right). \tag{14}$$

We now show that for any large enough $t$, queue 1 can not hold many more packets than the non-deviating queues. Define

$$\phi_t^+ := \left(Q_t^1 - \max_{i\geq 3} Q_t^i - 2N\sqrt{t\ln(t)}\right)_+.$$

Once again, at every iteration where $\phi_t^+$ is strictly positive and Equation (12) holds, queue 1 holds the oldest packet and thus has priority on whichever server it chooses. This implies that for any large enough $t$, $\phi_t^+$ is a supermartingale. It also holds that for any $t \geq 0$, $|\phi_{t+1}^+ - \phi_t^+| \leq 2(N+1)$. Thus, with the same arguments used to prove Equation (13), almost surely,

$$\left(Q_t^1 - \max_{i\geq 3} Q_t^i\right)_+ = \tilde{\mathcal{O}}\left(\sqrt{t}\right). \tag{15}$$

With that at hand, we prove that for any large enough $k$, queue 1 does not get priority often over the other queues during the first stage of the $k^{\text{th}}$ window. For any $i \in \{2,\ldots,N/2\}$, pose:

$$\psi_\ell^i = \frac{1}{2}\left(Q_{\lceil \ell N\rceil}^{2i-1} + Q_{\lceil \ell N\rceil}^{2i}\right) - Q_{\lceil \ell N\rceil}^1 - \frac{2(N-d)}{N^3}(\lceil \ell N\rceil - W_{k-1})$$

For any $\ell$ s.t. $\{\lceil \ell N\rceil; \lceil (\ell+1)N-1\rceil\}$ is included in the first phase of a window, we have

$$\sum_{t=\lceil \ell N\rceil}^{\lceil (\ell+1)N\rceil-1} \mathbb{E}\left[C_t^1 \middle| \psi_{1:\ell}^+(i,j)\right] \geq \sum_{t=\lceil \ell N\rceil}^{\lceil (\ell+1)N\rceil-1} \mathbb{E}\left[S_i^t \mathbb{1}_{\{\text{queue 1 and only queue 1 picks server } i\}} \middle| \psi_{1:\ell}^+(i,j)\right]$$

$$\geq \frac{N-d}{N} + \frac{2(N-d)}{N^2}$$

as well as

$$\sum_{t=\lceil \ell N\rceil}^{\lceil (\ell+1)N\rceil-1} \mathbb{E}\left[\frac{1}{2}\left(C_t^{2i} + C_t^{2i-1}\right) \middle| \psi_{1:\ell}^+(i,j)\right] \leq \sum_{t=\lceil \ell N\rceil}^{\lceil (\ell+1)N\rceil-1} \mathbb{E}\left[\frac{1}{2} S_{i+t \pmod N}^t \middle| \psi_{1:\ell}^+(i,j)\right]$$

$$\leq \frac{N-d}{N}.$$

Those two inequalities imply:

$$\mathbb{E}[\psi_{\ell+1}^i | \psi_{1:\ell}^+(i,j)] = \psi_\ell^+(i,j) + \sum_{t=\lceil \ell N \rceil}^{\lceil (\ell+1)N \rceil - 1} \mathbb{E}\left[\frac{1}{2}(B_t^{2i} - B_t^{2i-1}) - B_t^1 \Big| \psi_{1:\ell}^+(i,j)\right]$$

$$- \sum_{t=\lceil \ell N \rceil}^{\lceil (\ell+1)N \rceil - 1} \mathbb{E}\left[\frac{1}{2}(C_t^{2i} - C_t^{2i-1}) - C_t^1 \Big| \psi_{1:\ell}^+(i,j)\right] - \frac{2(N-d)}{N^2}$$

$$\geq \psi_\ell^+(i,j).$$

Thus, for any $\ell$ s.t. $\{\lceil \ell N \rceil; \lceil (\ell+1)N - 1 \rceil\}$ is included in the first phase of a window, $\psi_\ell^i$ is a submartingale. Moreover, for any $\ell \geq 0$, $|\psi_{\ell+1}^i - \psi_\ell^i| \leq 3N$. Thus, by Azuma-Hoeffding's inequality, for any $\ell$ s.t.$\{\lceil \ell N \rceil; \lceil (\ell+1)N - 1 \rceil\} \subset [W_{k-1}, W_{k-1} + \alpha w_k]$,

$$\Pr\left(\psi_\ell^i - \psi_{W_k}^i \leq -6N\sqrt{\ell N \ln(\ell N)}\right) \leq \frac{1}{(\ell N)^2}.$$

Borel-Cantelli's lemma implies, that for any large enough $\ell$ s.t.$\{\lceil \ell N \rceil; \lceil (\ell+1)N - 1 \rceil\} \subset [W_{k-1}, W_{k-1} + \alpha w_k]$, almost surely:

$$\psi_\ell^i \geq \psi_{W_k}^i - 6N\sqrt{\ell N \ln(\ell N)}.$$

This and Equation (15) applied at $t = W_k$, imply that for any large enough $k$, for any $t \in [W_{k-1}, W_{k-1} + \alpha w_k]$,

$$\frac{1}{2}\left(Q_t^{2i-1} + Q_t^{2i}\right) \geq Q_{\lceil \ell N \rceil}^1 + \frac{2(N-d)}{N^3}(t - W_{k-1}) + \psi_{W_k}^i - \tilde{O}(\sqrt{t})$$

$$\geq Q_{\lceil \ell N \rceil}^1 + \frac{2(N-d)}{N^3}(t - W_{k-1}) - \tilde{O}(w_k^{3/4}).$$

This and Equation (12) imply that during the first stage of the time window, queue 1 holds younger packets than any other queues $i \geq 3$ after at most $\tilde{\mathcal{O}}(w_k^{3/4})$ iterations.

By Chernoff bound and the Borel-Cantelli lemma again, for any large enough $k$, almost surely, the number of packets queue 1 clears during the first stage of the $k^{\text{th}}$ window on servers where it does not collide with other queues is:

$$\sum_{t=W_{k-1}+1}^{W_{k-1}+\alpha w_k} \sum_{i=1}^N S_i^t \mathbb{1}_{\{\text{queue 1 and only queue 1 picks server } i\}} = \left(\frac{1}{2} + \frac{1}{N}\right)\frac{2(N-d)}{N^2}\alpha w_k + \tilde{\mathcal{O}}\left(\sqrt{w_k}\right).$$

Since we have shown that for any large enough $k$, almost surely, queue 1 does not have priority over the other queues after at most $\tilde{\mathcal{O}}(w_k^{3/4})$ iterations, for any large enough $k$, almost surely, the number of packets queue 1 clears of the first stage of the $k^{\text{th}}$ window is

$$\left(\frac{1}{2} + \frac{1}{N}\right)\frac{2(N-d)}{N^2}\alpha w_k + \tilde{\mathcal{O}}\left(w_k^{3/4}\right).$$

$\square$

We are now ready to prove Lemma 6.

*Proof.* By Chernoff bound and the Borel-Cantelli lemma, almost surely for any large enough $k$, the number of packets queue 1 clears during the second stage of the window on servers where it does not collide with other queues is:

$$\sum_{t=W_{k-1}+\alpha w_k}^{W_k-1} \sum_{i=1}^N S_i^t \mathbb{1}_{\{\text{queue 1 and only queue 1 picks server } i\}} = \frac{4(N-d)}{N^3}(1-\alpha)w_k + \tilde{\mathcal{O}}\left(\sqrt{w_k}\right). \quad (16)$$

Suppose that during the second stage of the window, queue 1 never gets priority over another queue. In that case, according to Equation (16) and Lemma 7, for any large enough $k$, almost surely, the total number of packets cleared by queue 1 during the time window is

$$\left(\frac{\alpha}{2} + \frac{2-\alpha}{N}\right) \frac{2(N-d)}{N^2} w_k + \tilde{\mathcal{O}}(w_k^{3/4}).$$

For any large enough $k$, if $\alpha \leq 1 - \frac{1}{N-1}$ this is smaller than the number of packets queue 1 would have cleared had it not deviated, according to Lemma 5.

On the other hand, suppose that queue gets priority over some other queue $i$ at some iteration $\tau$ of the second stage of the window. In that case, at that iteration, queue 1 holds the oldest packets, which, according to Equation (12), implies

$$Q_1^\tau > Q_i^\tau - \tilde{\mathcal{O}}(w_k^{3/4})$$

During the second stage of the window, for any $i \geq 3$, $\gamma_t^i := \left(Q_i^t - Q_1^t - 2N\sqrt{t\ln(t)}\right)_+$ is a supermartingale with bounded increments for any $t$ where Equation (12) holds for queues 1 and $i$. Indeed, in that case, if $\gamma_t^i$ is strictly positive, queue $i$ holds an older packet than queue 1, and thus, whether they collide or not, it has a higher probability to clear a packet than queue 1. Thus, by Azuma-Hoeffding and the Borel-cantelli lemma again, for any large enough $k$, almost surely,

$$Q_i^{W_k} - Q_1^{W_k} \leq Q_i^\tau - Q_1^\tau + \tilde{\mathcal{O}}(w_k^{3/4}).$$

Thus it holds that $Q_1^{W_k} \geq Q_i^{W_k} - \tilde{\mathcal{O}}(w_k^{3/4})$ for any $i \geq 2$. This and Equation (15) imply that all the queues clear approximately the same number of packets over those time windows for any large enough $k$ almost surely. Thus queue 1 clears

$$\left[(2-\alpha)(N-2) + (\alpha + \frac{4-2\alpha}{N})\right] \frac{(N-d)}{(N-1)N^2} w_k + \tilde{\mathcal{O}}\left(w_k^{3/4}\right)$$

packets almost surely, which again is smaller than the number of packets it would have cleared had it not deviated.

Thus, the deviating queue clears almost surely less packets by time window than it would have had it not deviated on all but finitely many of the time windows, which implies that it has no policy regret on all but finitely many of the time windows. $\square$

## D Proofs of Section 4

### D.1 Proof of Lemma 1

We want to show that if $\|(\hat{\lambda} - \lambda, \hat{\mu} - \mu)\|_\infty \leq c_1\Delta$, then

$$\|\phi(\hat{\lambda}, \hat{\mu}) - \phi(\lambda, \mu)\|_2 \leq \frac{c_2 K}{\Delta} \|(\hat{\lambda} - \lambda, \hat{\mu} - \mu)\|_\infty, \tag{17}$$

with the constants $c_1, c_2$ given in Lemma 1.

Recall that $\phi$ is defined as

$$\phi(\lambda, \mu) = \arg\min_{P \in \mathfrak{B}_K} f(P, \lambda, \mu),$$

where $\mathfrak{B}_K$ is the set of $K \times K$ doubly stochastic matrices and $f$ is defined as:

$$f(P, \lambda, \mu) := \max_{i \in [N]} -\ln(\sum_{j=1}^K P_{i,j}\mu_k - \lambda_i) + \frac{1}{2K}\|P\|_2^2$$

Let $P^*$ and $\hat{P}^*$ be the minimizers of $f$ with the respective parameters $(\lambda, \mu)$ and $(\hat{\lambda}, \hat{\mu})$. They are uniquely defined as $f$ is $\frac{1}{K}$ strongly convex.

As the property of Lemma 1 is symmetric, we can assume without loss of generality that $f(P^*, \lambda, \mu) \geq f(\hat{P}^*, \hat{\lambda}, \hat{\mu})$.

Given the definition of $\Delta$, we have the bound

$$-\ln(\Delta) + \frac{1}{2} \geq f(P^*, \lambda, \mu) \geq -\ln(\Delta).$$

The lower bound holds because the term in the $\ln$ is at most $\Delta$ for at least one $i$. For the upper bound, some matrix $P$ ensures that the term in the $\ln$ is at least $\Delta$ for all $i$ and $\|P\|_2^2 \leq K$.

Similarly for $\hat{P}^*$, it follows:

$$-\ln((1 - 2c_1)\Delta) + \frac{1}{2} \geq f(\hat{P}^*, \hat{\lambda}, \hat{\mu}) \geq -\ln((1 + 2c_1)\Delta).$$

As a consequence, it holds for any $i \in [N]$:

$$-\ln\left(\sum_{j=1}^{K} \hat{P}_{i,j}^* \hat{\mu}_j - \hat{\lambda}_i\right) \leq f(\hat{P}^*, \hat{\lambda}, \hat{\mu})$$

$$\leq -\ln((1 - 2c_1)\Delta/\sqrt{e})$$

$$\sum_{j=1}^{K} \hat{P}_{i,j}^* \hat{\mu}_j - \hat{\lambda}_i \geq (1 - 2c_1)\Delta/\sqrt{e}.$$

Note that for any $i \in [N]$,

$$\sum_{j=1}^{K} \hat{P}_{i,j}^* \hat{\mu}_j - \hat{\lambda}_i \leq \sum_{j=1}^{K} \hat{P}_{i,j}^* \mu_j - \lambda_i + 2\|(\hat{\lambda} - \lambda, \hat{\mu} - \mu)\|_\infty.$$

It then yields the second point of Lemma 1:

$$\sum_{j=1}^{K} \hat{P}_{i,j}^* \mu_j - \lambda_i \geq \left((1 - 2c_1)/\sqrt{e} - 2c_1\right)\Delta$$

Moreover, it follows

$$-\ln\left(\sum_{j=1}^{K} \hat{P}_{i,j}^* \hat{\mu}_j - \hat{\lambda}_i\right) \geq -\ln\left(\sum_{j=1}^{K} \hat{P}_{i,j}^* \mu_j - \lambda_i\right) - \ln\left(1 + \frac{2\|(\hat{\lambda} - \lambda, \hat{\mu} - \mu)\|_\infty}{\sum_{j=1}^{K} \hat{P}_{i,j}^* \mu_j - \lambda_i}\right)$$

$$\geq -\ln\left(\sum_{j=1}^{K} \hat{P}_{i,j}^* \mu_j - \lambda_i\right) - \frac{2\|(\hat{\lambda} - \lambda, \hat{\mu} - \mu)\|_\infty}{\left((1 - 2c_1)/\sqrt{e} - 2c_1\right)\Delta}$$

Recall that for a $\frac{1}{K}$-strongly convex function $g$ of global minimum $x^*$ and any $x$:

$$\|x - x^*\|_2 \leq 2K(g(x) - g(x^*))$$

As a consequence, it follows:

$$f(\hat{P}^*, \hat{\lambda}, \hat{\mu}) \geq f(\hat{P}^*, \lambda, \mu) - \frac{2\|(\hat{\lambda} - \lambda, \hat{\mu} - \mu)\|_\infty}{\left((1 - 2c_1)/\sqrt{e} - 2c_1\right)\Delta}$$

$$\geq f(P^*, \lambda, \mu) - \frac{2\|(\hat{\lambda} - \lambda, \hat{\mu} - \mu)\|_\infty}{\left((1 - 2c_1)/\sqrt{e} - 2c_1\right)\Delta} + \frac{1}{2K}\|P^* - \hat{P}^*\|_2.$$

Equation (17) then follows.

## D.2 Proof of Lemma 2

The coefficient $C_{i,j}$ is replaced by $+\infty$ as soon as the whole weight $P_{i,j}$ is exhausted. Thanks to this, the HUNGARIAN algorithm does return a perfect matching with respect to the bipartite graph with edges $(i, j)$ where there remains some weight for $P_{i,j}$. Because of this, it can be shown following the usual proof of Birkhoff algorithm [Birkhoff, 1946] that the sequence $(z[j], P[j])$ is indeed of length at most $K^2 - K + 1$ and is a valid decomposition of $P$.

Now assume that $\prec_C$ is a total order. At each iteration $j$ of HUNGARIAN algorithm, denote $\tilde{P}^j = P - \sum_{s=1}^{j-1} z[s]P[s]$ the remaining weights to attribute.

Let $l_j$ be such that $P[j] = P_{l_j}$ for any iteration $j$ of HUNGARIAN algorithm.

It can now be shown by induction that

$$\tilde{P}^j = P - \sum_{l=1}^{l_j} z_l(P)P_l.$$

where $z_l(P)$ are defined by Equation (5). Indeed, by definition

$$\tilde{P}^{j+1} = \tilde{P}^j - z[j+1]P[j+1]$$
$$= \tilde{P}^j - z[j+1]P_{l_{j+1}}$$

The HUNGARIAN algorithm returns the minimal cost matching with respect to the modified cost matrix $C$ where the coefficients $i, k$ such that $\tilde{P}^j_{i,k} = 0$ are replaced by $+\infty$. Thanks to this, $P_{l_{j+1}}$ is the minimal cost permutation matrix $P_l$ (for $\prec_C$) such that $\tilde{P}^j_{i,k} > 0$ for all $(i, k) \in \text{supp}(P_l)$.

This means that for any $l < l_{j+1}$

$$\min_{(i,k)\in\text{supp}(P_l)} (\tilde{P}^j)_{i,k} = 0.$$

Using the induction hypothesis, this implies that $z_l(P) = 0$ for any $l_j < l < l_{j+1}$. And finally, this also implies that $z_{l_{j+1}}(P) = z[j+1]$.

This finally concludes the proof as $\tilde{P}^j = 0$ after the last iteration.

## D.3 Proof of Lemma 3

For $z$ and $z'$ the respective decompositions of $P$ and $P'$ defined in Lemma 2, then

$$\int_0^1 \mathbb{1}\left(\psi(P)(\omega) \neq \psi(P')(\omega)\right) d\omega = \mathbb{P}_{\omega\sim U(0,1)}\left(\psi(P)(\omega) \neq \psi(P')(\omega)\right).$$

In the following, note $A = \psi(P)$ and $A' = \psi(P')$. It follows for $P_n$ defined as in Lemma 2

$$\int_0^1 \mathbb{1}\left(\psi(P)(\omega) \neq \psi(P')(\omega)\right) d\omega = \sum_{n=1}^{K!} \mathbb{P}(A = P_n \text{ and } A' \neq P_n)$$

$$= \frac{1}{2}\sum_{n=1}^{K!} \mathbb{P}(A = P_n \text{ and } A' \neq P_n) + \frac{1}{2}\sum_{n=1}^{K!} \mathbb{P}(A' = P_n \text{ and } A \neq P_n)$$

$$= \frac{1}{2}\sum_{n=1}^{K!} \text{vol}\left(\left[\sum_{j=1}^{n-1} z_j(P), \sum_{j=1}^{n} z_j(P)\right] \ominus \left[\sum_{j=1}^{n-1} z_j(P'), \sum_{j=1}^{n} z_j(P')\right]\right),$$

where vol denotes the volume of a set and $A \ominus B = (A \setminus B) \cup (B \setminus A)$ is the symmetric difference of $A$ and $B$. The last equality comes from the expression of $\psi$ with respect to the coefficients $z_j(P)$, thanks to Lemma 2.

It is easy to show that
$$\text{vol}\left([a,b]\ominus[c,d]\right)\leq\left(|c-a|+|d-b|\right)\mathbb{1}\left(b>a \text{ or } c>d\right).$$
The previous equality then leads to

$$\int_0^1\mathbb{1}\left(\psi(P)(\omega)\neq\psi(P')(\omega)\right)\mathrm{d}\omega\leq\frac{1}{2}\sum_{n=1}^{K!}\left(\left|\sum_{j=1}^{n-1}z_j(P)-z_j(P')\right|+\left|\sum_{j=1}^{n}z_j(P)-z_j(P')\right|\right)\mathbb{1}\left(z_n(P)+z_n(P')>0\right)$$

$$\leq\sum_{n=1}^{K!}\left|\sum_{j=1}^{n}z_j(P)-z_j(P')\right|\mathbb{1}\left(z_n(P)+z_n(P')>0\right).\qquad(18)$$

The last inequality holds because $\sum_{j=1}^{k}z_j(P)-z_j(P')$ is counted twice when $z_k(P)+z_k(P')$ is positive: when $n=k$ and for the next $n$ such that the elements are counted in the sum.

Thanks to Lemma 2, only $2(K^2-K+1)$ elements $z_j(P)$ and $z_j(P')$ are non-zero. Let $k_n$ be the index of the $n$-th non-zero element of $(z_s(P)+z_s(P'))_{1\leq s\leq K!}$. Note that $z_s(P')$ can be non-zero while $z_s(P)$ is zero (or conversely). Let also

$$(i_{k_n},j_{k_n})\in\operatorname*{arg\,min}_{(i,j)\in\text{supp}(P_{k_n})}P_{i,j}-\sum_{l<k_n}z_l(P)\mathbb{1}\left((i,j)\in\text{supp}(P_{k_n})\right),$$

$$(i'_{k_n},j'_{k_n})\in\operatorname*{arg\,min}_{(i,j)\in\text{supp}(P_{k_n})}P'_{i,j}-\sum_{l<k_n}z_l(P')\mathbb{1}\left((i,j)\in\text{supp}(P_{k_n})\right).$$

It then comes, thanks to Lemma 2

$$z_{k_n}(P)-z_{k_n}(P')\leq P_{i'_{k_n},j'_{k_n}}-P'_{i'_{k_n},j'k_n}-\sum_{l<k_n}(z_l(P)-z_l(P'))\mathbb{1}\left((i'_{k_n},j'_{k_n})\in\text{supp}(P_{k_n})\right)$$

$$\leq P_{i'_{k_n},j'_{k_n}}-P'_{i'_{k_n},j'_{k_n}}-\sum_{l<n}(z_{k_l}(P)-z_{k_l}(P'))\mathbb{1}\left((i'_{k_n},j'_{k_n})\in\text{supp}(P_{k_n})\right)$$

The second inequality holds, because for $l'\notin\{k_l\mid l<2K^2\}$, the term in the sum is zero by definition of the sequence $k_l$.

A similar inequality holds for $z_{k_n}(P')-z_{k_n}(P)$, which leads to
$$|z_{k_n}(P)-z_{k_n}(P')|\leq\|P-P'\|_\infty+\sum_{l<n}|z_{k_l}(P)-z_{k_l}(P')|.$$

By induction, it thus holds
$$|z_{k_n}(P)-z_{k_n}(P')|\leq 2^{n-1}\|P-P'\|_\infty.$$

We finally conclude using Equation (18)

$$\int_0^1\mathbb{1}\left(\psi(P)(\omega)\neq\psi(P')(\omega)\right)\mathrm{d}\omega\leq\sum_{n=1}^{K!}\left|\sum_{j=1}^{n}z_j(P)-z_j(P')\right|\mathbb{1}\left(z_n(P)+z_n(P')>0\right)$$

$$\leq\sum_{n=1}^{2(K^2-K+1)-1}\left|\sum_{j=1}^{k_n}z_j(P)-z_j(P')\right|$$

$$\leq\sum_{n=1}^{2(K^2-K+1)-1}\left|\sum_{l=1}^{n}z_{k_l}(P)-z_{k_l}(P')\right|$$

$$\leq\sum_{n=1}^{2(K^2-K+1)-1}\sum_{j=1}^{n}2^{j-1}\|P-P'\|_\infty$$

$$\leq 2^{2(K^2-K+1)}\|P-P'\|_\infty.$$

In the fourth inequality, the $2(K^2-K+1)$-th term of the sum is ignored. It is indeed 0 as $z$ and $z'$ both sum to 1.

## D.4   Proof of Theorem 5

First recall below a useful version of Chernoff bound.

**Lemma 8.** *For any independent variables $X_1, \ldots, X_n$ in $[0,1]$ and $\delta \in (0,1)$,*

$$\mathbb{P}\left(\sum_{i=1}^{n} X_i \leq (1-\delta) \sum_{i=1}^{n} \mathbb{E}[X_i]\right) \leq e^{-\frac{\delta^2 \sum_{i=1}^{n} \mathbb{E}[X_i]}{2}}.$$

We now prove the following concentration lemma.

**Lemma 9.** *For any time $t \geq (N+K)^5$ and $\varepsilon \in (0, \frac{1}{4})$,*

$$\mathbb{P}\left(|\hat{\mu}_k^i(t) - \mu_k| \geq \varepsilon\right) \leq 3\exp\left(-\lambda_i \left(t^{\frac{4}{5}} - 1\right)\varepsilon^2\right)$$

$$\mathbb{P}\left(|\hat{\lambda}_j^i(t) - \lambda_j| \geq \varepsilon\right) \leq 6\exp\left(-\lambda_i K\bar{\mu}\frac{t^{\frac{4}{5}} - 1}{145}\varepsilon^2\right).$$

*Proof.*
**Concentration for $\hat{\mu}$.** Consider agent $i$ in the following and denote by $N_k(t)$ the number of *exploratory pulls* of this agent on server $k$ at time $t$. By definition, the probability to proceed to an exploratory pull on the server $k$ at round $t$ is at least $\lambda_i \min(t^{-\frac{1}{5}}, \frac{1}{N+K})$. The term $\lambda_i$ here appears as a pull is guaranteed if a packet appeared at the current time step. Yet the number of exploratory pulls might be much larger in practice as queues should accumulate a large number of uncleared packets at the beginning.

For $t \geq (N+K)^5$, it holds:

$$\sum_{n=1}^{t} \min(n^{-\frac{1}{5}}, \frac{1}{N+K}) = \sum_{n=1}^{(N+K)^5} \frac{1}{N+K} + \sum_{n=(N+K)^5+1}^{t} n^{-\frac{1}{5}}$$

$$\geq (N+K)^4 + \int_{(N+K)^5}^{t} x^{-\frac{1}{5}}\mathrm{d}x - 1$$

$$\geq \frac{1}{4}\left(5t^{\frac{4}{5}} - (N+K)^4 - 4\right)$$

$$\geq t^{\frac{4}{5}} - 1.$$

Lemma 8 then gives for $N_k(t)$:

$$\mathbb{P}\left(N_k(t) \leq (1-\delta)\mathbb{E}[N_k(t)]\right) \leq \exp\left(-\frac{\delta^2 \mathbb{E}[N_k(t)]}{2}\right)$$

$$\mathbb{P}\left(N_k(t) \leq (1-\delta)\lambda_i\left(t^{\frac{4}{5}} - 1\right)\right) \leq \exp\left(-\frac{\lambda_i \delta^2\left(t^{\frac{4}{5}} - 1\right)}{2}\right).$$

Which leads for $\delta = \frac{1}{2}$ to

$$\mathbb{P}\left(N_k(t) \leq \frac{\lambda_i}{2}\left(t^{\frac{4}{5}} - 1\right)\right) \leq \exp\left(-\lambda_i \frac{t^{\frac{4}{5}} - 1}{8}\right). \tag{19}$$

The number of exploratory pulls and the observations on the server $k$ are independent. Thanks to this, Hoeffding's inequality can be directly used as follows

$$\mathbb{P}\left(|\hat{\mu}_k^i(t) - \mu_k| \geq \varepsilon \mid N_k(t)\right) \leq 2\exp\left(-2N_k(t)\varepsilon^2\right).$$

Using Equation (19) now gives the first concentration inequality for $\varepsilon \leq \frac{1}{4} \leq \frac{1}{2\sqrt{2}}$:

$$\mathbb{P}\left(|\hat{\mu}_k^i(t) - \mu_k| \geq \varepsilon\right) \leq 2\exp\left(-\lambda_i\left(t^{\frac{4}{5}} - 1\right)\varepsilon^2\right) + \exp\left(-\lambda_i \frac{t^{\frac{4}{5}} - 1}{8}\right)$$

$$\leq 3\exp\left(-\lambda_i\left(t^{\frac{4}{5}} - 1\right)\varepsilon^2\right).$$

**Concentration for $\hat{\lambda}$.** Consider agent $i$ in the following. First show a concentration inequality for $\tilde{\mu}$. Denote by $N(t)$ the total number of exploratory pulls on servers proceeded by player $i$ at round $t$, i.e., $N(t) = \sum_{k=1}^{K} N_k(t)$. Similarly to Equation (19), it can be shown that

$$\mathbb{P}\left( N(t) \le \lambda_i K \frac{t^{\frac{4}{5}} - 1}{2} \right) \le \exp\left( -\lambda_i K \frac{t^{\frac{4}{5}} - 1}{8} \right).$$

Lemma 8 then gives for $\delta \in (0,1)$:

$$\mathbb{P}\left( |\tilde{\mu} - \bar{\mu}| \ge \delta \bar{\mu} \right) \le 2\exp\left( -\lambda_i K \delta^2 \bar{\mu} \frac{t^{\frac{4}{5}} - 1}{8} \right) + \exp\left( -\lambda_i K \frac{t^{\frac{4}{5}} - 1}{8} \right)$$

$$\le 3\exp\left( -\lambda_i K \delta^2 \bar{\mu} \frac{t^{\frac{4}{5}} - 1}{8} \right).$$

Note that $|\tilde{\mu} - \bar{\mu}| \le \delta\bar{\mu}$ implies $\left| \frac{1}{\tilde{\mu}} - \frac{1}{\bar{\mu}} \right| \le \frac{\delta}{(1-\delta)\bar{\mu}}$. So this gives the following inequality:

$$\mathbb{P}\left( \left| \frac{1}{\tilde{\mu}} - \frac{1}{\bar{\mu}} \right| \ge \frac{\delta}{(1-\delta)\bar{\mu}} \right) \le 3\exp\left( -\lambda_i K \delta^2 \bar{\mu} \frac{t^{\frac{4}{5}} - 1}{8} \right). \tag{20}$$

A concentration bound on $\hat{S}_j^i$ can be shown similarly for any $\delta \in (0,1)$

$$\mathbb{P}\left( \left| \hat{S}_j^i(t) - (1 - \frac{\lambda_j}{2})\bar{\mu} \right| \ge \delta\bar{\mu} \right) \le 3\exp\left( -\lambda_i K \delta^2 \bar{\mu} \frac{t^{\frac{4}{5}} - 1}{8} \right). \tag{21}$$

Now recall that the estimate of $\lambda_j$ is defined by $\hat{\lambda}_j = 2 - \frac{2\hat{S}_j^i}{\tilde{\mu}}$. We then have the following identity:

$$\hat{\lambda}_j - \lambda_j = 2(\frac{1}{\tilde{\mu}} - \frac{1}{\bar{\mu}})\hat{S}_j^i + \frac{2}{\bar{\mu}}\left( (1 - \frac{\lambda_j}{2})\bar{\mu} - \hat{S}_j^i \right).$$

Since $\hat{S}_j \in [0,1]$, it yields for $\varepsilon \le \frac{1}{4}$ and $x \in (0,1)$:

$$\mathbb{P}\left( \left| \hat{\lambda}_j^i(t) - \lambda_j \right| \ge \varepsilon \right) \le \mathbb{P}\left( \left| 2(\frac{1}{\tilde{\mu}} - \frac{1}{\bar{\mu}})\hat{S}_j^i \right| \ge x\varepsilon \text{ or } \left| \frac{2}{\bar{\mu}}\left( (1 - \frac{\lambda_j}{2})\bar{\mu} - \hat{S}_j \right) \right| \ge (1-x)\varepsilon \right)$$

$$\le \mathbb{P}\left( \left| \frac{1}{\tilde{\mu}} - \frac{1}{\bar{\mu}} \right| \ge \frac{x\varepsilon}{2\hat{S}_j^i} \mid \hat{S}_j^i \le (1 + \frac{1-x}{8})\bar{\mu} \right) + \mathbb{P}\left( \left| \hat{S}_j^i(t) - (1 - \frac{\lambda_j}{2})\bar{\mu} \right| \ge \frac{(1-x)\bar{\mu}\varepsilon}{2} \right)$$

$$\le \mathbb{P}\left( \left| \frac{1}{\tilde{\mu}} - \frac{1}{\bar{\mu}} \right| \ge \frac{\delta}{(1-\delta)\bar{\mu}} \text{ for } \delta = \frac{4x\varepsilon}{9} \right) + \mathbb{P}\left( \left| \hat{S}_j^i(t) - (1 - \frac{\lambda_j}{2})\bar{\mu} \right| \ge \frac{(1-x)\bar{\mu}\varepsilon}{2} \right)$$

Taking $x = \frac{9}{17}$ leads to $\frac{4x}{9} = \frac{1-x}{2}$ and thus, using Equations (20) and (21):

$$\mathbb{P}\left( \left| \hat{\lambda}_j^i(t) - \lambda_j \right| \ge \varepsilon \right) \le 6\exp\left( -\lambda_i K \left( \frac{8}{17} \right)^2 \bar{\mu} \frac{t^{\frac{4}{5}} - 1}{32} \varepsilon^2 \right)$$

$$\le 6\exp\left( -\lambda_i K \bar{\mu} \frac{t^{\frac{4}{5}} - 1}{145} \varepsilon^2 \right)$$

$\square$

In the following, let $c_1 = 0.1$ and $c_2 = \frac{4}{(1-2c_1)/\sqrt{e} - 2c_1} \approx 14$. For a problem instance, let the good event $\mathcal{E}_t$ at time $t$ be defined as

$$\mathcal{E}_t := \left\{ \|(\hat{\lambda}^i - \lambda, \hat{\mu}^i - \mu)\|_\infty \leq \frac{0.1\Delta^2}{2c_2 2^{2K^2} KN}, \ \forall i \in [N] \right\}.$$

As $\Delta$ is smaller than 1, the right hand term in the definition of $\mathcal{E}_t$ is smaller than $c_1\Delta$. Thanks to Lemmas 1 and 3, $\mathcal{E}_t$ then guarantees that any player will collide with another player with probability at most $0.1\Delta$, i.e., , $\forall i \in [N]$,

$$\mathbb{P}_{\omega \sim \mathcal{U}(0,1)} \left( \exists j \in [N], \hat{A}_t^i(\omega) \neq \hat{A}_t^j(\omega) \mid \mathcal{E}_t \right) \leq 0.1\Delta.$$

Moreover, thanks to Lemma 1, under $\mathcal{E}_t$,

$$\lambda_i \leq \sum_{k=1}^{K} \hat{P}_{i,k}\mu_k - \left( \frac{1 - 2c_1}{\sqrt{e}} - 2c_1 \right) \Delta.$$

These two last inequalities lead to the following lemma.

**Lemma 10.** *For $t \geq \frac{2^5 K^5}{0.08^5 \Delta^5}$, denote by $\mathcal{H}_t$ the history of observations up to round $t$. Then*

$$\mathbb{E}\left[ S_t^i \mid \mathcal{E}_t, \mathcal{H}_t \right] \geq \lambda_i + 0.1\Delta.$$

*Proof.* This is a direct consequence of the following decomposition:

$$\mathbb{E}\left[ S_t^i \mid \mathcal{E}_t, \mathcal{H}_t \right] \geq \overbrace{(1 - (N+K)t^{-\frac{1}{5}})}^{\text{proba to exploit}} \left( \overbrace{\hat{P}_{i,k}\mu_k}^{\text{proba to clear}} - \overbrace{\mathbb{P}\left( \exists j \in [N], \hat{A}_t^i(\omega) \neq \hat{A}_t^j(\omega) \exists \mid \mathcal{E}_t \right)}^{\text{collision proba}} \right)$$

$$\geq (1 - (N+K)t^{-\frac{1}{5}})(\lambda_i + \left( \frac{1 - 2c_1}{\sqrt{e}} - 2c_1 \right)\Delta - 0.1\Delta)$$

$$\geq (1 - (N+K)t^{-\frac{1}{5}})(\lambda_i + 0.18\Delta).$$

The last inequality is given by $c_1 = 0.1$ and it leads to

$$\mathbb{E}\left[ S_t^i \mid \mathcal{E}_t, \mathcal{H}_t \right] \geq \lambda_i + 0.18\Delta - (N+K)t^{-\frac{1}{5}}.$$

For $t \geq \frac{2^5 K^5}{0.08^5 \Delta^5}$, the last term is smaller than $0.08\Delta$, giving Lemma 10. $\qquad \square$

Define the stopping time

$$\tau := \min\left\{ t \geq \frac{2^5 K^5}{0.08^5 \Delta^5} \mid \forall t \geq s, \mathcal{E}_s \text{ holds} \right\}. \tag{22}$$

**Lemma 11.** *For any integer $r \geq 1$,*

$$\mathbb{E}[\tau^r] = \mathcal{O}\left( KN \left( \frac{N^{\frac{5}{2}} K^{\frac{5}{2}} 2^{5K^2}}{(\min(1, K\bar{\mu})\lambda)^{\frac{5}{4}} \Delta^5} \right)^r \right),$$

*where the $\mathcal{O}$ notation hides constant factors that only depend on $r$.*

*Proof.* Define for this proof $t_0 = \left\lceil \frac{2^5 K^5}{0.08^5 \Delta^5} \right\rceil$. By definition, if $\mathcal{E}_t$ does not hold for $t > t_0$, then $\tau \geq t$. As a consequence, for any $t > t_0$ and thanks to Lemma 9:

$$\mathbb{P}(\tau \geq t) \leq \mathbb{P}(\neg\mathcal{E}_t)$$

$$\leq (3eKN + 6eN^2) \exp\left( -ct^{\frac{4}{5}} \right),$$

where $c = c_0 \frac{\min(1, K\bar{\mu})\lambda\Delta^4}{N^2 K^2 2^{4K^2}}$ for some universal constant $c_0 \leq 1$.

We can now bound the moments of $\tau$:

$$\mathbb{E}[\tau^r] = r \int_0^\infty t^{r-1} \mathbb{P}\left(\tau \geq t\right) \mathrm{d}t$$

$$\leq t_0^r + (3eKN + 6eN^2)r \int_0^\infty t^{r-1} e^{-ct^{\frac{4}{5}}} \mathrm{d}t.$$

Using the change of variable $u = ct^{\frac{4}{5}}$, it can be shown that

$$\int_0^\infty t^{r-1} e^{-ct^{\frac{4}{5}}} \mathrm{d}t = \frac{5}{4} c^{-\frac{5r}{4}} \Gamma\left(\frac{5r}{4}\right),$$

where $\Gamma$ denotes the Gamma function. It finally allows to conclude:

$$\mathbb{E}[\tau^r] = \mathcal{O}\left(\frac{K^{5r}}{\Delta^{5r}} + KNc^{-\frac{5r}{4}}\right)$$

$$= \mathcal{O}\left(KN\left(\frac{N^{\frac{5}{2}} K^{\frac{5}{2}} 2^{5K^2}}{(\min(1, K\bar{\mu})\lambda)^{\frac{5}{4}} \Delta^5}\right)^r\right).$$

$\square$

Let $X_t$ be a random walk biased towards $0$ with the following transition probabilities:

$$\mathbb{P}(X_{t+1} = X_t + 1) = p, \ \mathbb{P}(X_{t+1} = X_t - 1 | X_t > 0) = q,$$
$$\mathbb{P}(X_{t+1} = X_t | X_t > 0) = 1 - p - q, \ \mathbb{P}(X_{t+1} = X_t | X_t = 0) = 1 - p, \tag{23}$$

and $X_0 = 0$.

**Lemma 12.** *The non-asymptotic moments of the random walk defined by Equation* (23) *are bounded. For any $t > 0, r > 0$:*

$$\mathbb{E}\left[(X_t)^r\right] \leq \frac{r!}{(\ln(q/p))^r}.$$

*Proof*: Let $\pi$ be the stationary distribution of the random walk. It verifies the following system of equations:

$$\begin{cases} \pi(z) = p\pi(z-1) + q\pi(z+1) + (1-p-q)\pi(z), \ \forall z > 0 \\ \pi(0) = (1-p)\pi(0) + q\pi(1) \\ \sum \pi(z) = 1 \end{cases}$$

which gives:

$$\pi(z) = \frac{q-p}{q}\left(\frac{p}{q}\right)^z.$$

Equivalently, $\pi(z) = \mathbb{P}(\lfloor Y \rfloor = z)$ with Y an exponential random variable of parameter $\ln(q/p)$. This gives:

$$\mathbb{E}_{X \sim \pi}\left[(X)^r\right] \leq \frac{r!}{(\ln(q/p))^r}.$$

Let $\tilde{X}_t$ be the random walk with the same transition probabilities as $X_t$ and $\tilde{X}_0 \sim \pi$. For any $t > 0$, $\tilde{X}_t \sim \pi$. Moreover, for any $t > 0$, $\tilde{X}_t$ stochasticaly dominates $X_t$, which terminates the proof. $\square$

*Proof of Theorem 5.* For $\tau$ the stopping time defined by Equation (22), Lemma 11 bounds its moments as follows

$$\mathbb{E}[\tau^r] = \mathcal{O}\left(KN\left(\frac{N^{\frac{5}{2}} K^{\frac{5}{2}} 2^{5K^2}}{(\min(1, K\bar{\mu})\lambda)^{\frac{5}{4}} \Delta^5}\right)^r\right).$$

Let

$$p_i = \lambda_i(1 - \lambda_i - 0.1\Delta) \text{ and } q_i = (\lambda_i + 0.1\Delta)(1 - \lambda_i).$$

Let $X_t^i$ be the random walk biased towards 0 with parameters $p_i$ and $q_i$, with $X_t^i = 0$ for any $t \leq 0$. According to Lemma 10, past time $\tau$, $Q_t^i$ is stochastically dominated by the random process $\tau + X_{t-\tau}^i$. Thus, for any $t > 0$, for any $r > 0$

$$\mathbb{E}[(Q_i^t)^r] \leq \max(1, 2^{r-1}) \left( \mathbb{E}[\tau^r] + \mathbb{E}[(X_{t-\tau}^i)^r] \right)$$

$$= \mathcal{O}\left( KN \left( \frac{N^{\frac{5}{2}} K^{\frac{5}{2}} 2^{5K^2}}{(\min(1, K\bar{\mu})\underline{\lambda})^{\frac{5}{4}} \Delta^5} \right)^r + \frac{1}{\ln(q_i/p_i)^r} \right)$$

$$= \mathcal{O}\left( KN \left( \frac{N^{\frac{5}{2}} K^{\frac{5}{2}} 2^{5K^2}}{(\min(1, K\bar{\mu})\underline{\lambda})^{\frac{5}{4}} \Delta^5} \right)^r + \Delta^{-r} \right)$$

$$= \mathcal{O}\left( KN \left( \frac{N^{\frac{5}{2}} K^{\frac{5}{2}} 2^{5K^2}}{(\min(1, K\bar{\mu})\underline{\lambda})^{\frac{5}{4}} \Delta^5} \right)^r \right).$$

$\square$