# OpenReview forum: "Decentralized Learning in Online Queuing Systems"
_NeurIPS.cc/2021/Conference — NeurIPS 2021 Spotlight_

### Official Review · Reviewer_E9jT · 2021-07-06

**Rating:** 7
**Confidence:** 4

**Summary:**

The paper proposed a distributed scheduling algorithm for N queues and K servers, with heterogeneous arrival rates for the queues and service rates for the servers. The rates are unknown to the players (each player is a queue). Each queue to which server (if any) to send a packet. A server that receives multiple packets attempts to serve the oldest one and ignores the others (e.g., sends them back). In the proposed algorithm, players explore the arrival and service rates on the fly. To exploit, players use their local estimated rates to compute a dominant mapping between queues and servers based. Then players use Birkhoff von Neumann decomposition to obtain a doubly stochastic matrix of the random strategies, which gives the dominant mapping on average. It is proved that the proposed algorithm is strongly stable for when the slack eta is greater than one ("sum of service rate is larger than the sum of arrival rates").

**Limitations And Societal Impact:**

There is no negative societal impact since the paper is focused on the theory.

**Main Review:**

The paper is exciting and creative, solves a relevant problem, and is well-written. The proofs are sound and easy to follow. I recommend accepting the paper. There are only several issues that if resolved (or explained) can make the paper stronger:

1) The model needs to be discussed and motivated in more detail. Why is this structure where packets are sent back and forth to servers natural or efficient? is there any simple  *distributed* mechanism that implements that without actually submitting the packets? in what application would this be the case? a stronger model takes into account collisions between queues that send packets to the same server. How much more difficult is this model and why? (the exploitation avoids collisions already, doesn't it?). Also, reading lines 102-104 it looks like a queue has to send a packet if it's not empty. Is this true? seems like an arbitrary limitation.


2) That's of course subjective, but I didn't find Section 3 very interesting and it's not clear to me if it's relevant or delivers any strong point. The policy regret is only used in this section. As much as I understand, nobody proposed the policy regret as an alternative for queueing systems, so it's not very shocking that no-policy regret algorithms don't work. I do get why no policy regret is more suitable for queueing than simple regret. However, Proposition 1 doesn't even show that no-policy regret algorithms don't work, but that a very specific one doesn't work, that uses a lot of synchronization between the players. This is indeed a counterexample that shows that not *all* no policy regret algorithms work, but this sounds like a weak and esoteric point. It's possible that a smarter, less contrived no policy regret algorithm can have stability guarantees. It's probably hard to design and analyze such an algorithm, and it of course beyond the scope of this paper, but the result here doesn't say much about this possibility. Proposition 1 doesn't build the "case for a cooperative algorithm", since one can easily give a counterexample of a cooperative algorithm that doesn't work as well. Would this build the case against cooperative algorithms? I suggest moving this section to an appendix and tighten the language about what this result shows. The paper is exciting enough without this section, which delays the main point. The proposed cooperative algorithm seems like a breakthrough and it doesn't take away much from it if a no policy regret algorithm that can do the job exists out there.  Some technical questions:

a) How similar is the counterexample here to that used to prove Theorem 3?

b) it looks like T needs to be infinity for Proposition 1 to make sense, but the regret definition uses finite T. Then, saying that the policy regret is o(w_k) on all but finitely many windows is a bit cumbersome. Can't you say that the overall regret is sublinear?


3) The common randomness assumption seems reasonable to me, but it needs more discussion. The reader needs to be convinced that no hidden coordination is assumed so initializing the players with a "long random sequence" (seed) is enough. Consider discussing that here and there near the places where the common randomness is used.


4) The modification of the dominant mapping problem in (3) is nice, but I'm not sure if it's necessary. The cost is the assumption in Remark 3 that exact computation of a non-smooth problem is possible. This is perhaps the main weakness of the paper. Looking at the proof, I think there's some hope that working with the usual dominant mapping is possible. Line 272 says that the problem is that sorting the rates is non-continuous. Lines 255-257 imply the same (which might be an overstatement, by the way). However, the good event \mathcal(E_t) happens infinitely often, as the proof shows, for any estimation error for the rates. When this error is smaller than the minimal gap between the rates, all players agree on the ranking/sorting of the rates. Doesn't this allow for the usual dominant mapping computation? are there any other issues that cannot be easily solved? this needs to be at least discussed in more detail since the question naturally raises and currently the paper doesn't justify the modification well enough. In fact, Remark 3 is suggesting something very similar: that if the approximation error is small, there is no impact on the stability. The paper is strong enough already, but if such an improvement is possible, I think it's a small price to pay to make the paper even stronger.

5) A conclusion section can be nice. Aren't there any interesting insights or future directions to pursue? I definitely think there are plenty.



Minor Comments:

1) Line 117: "real number such that" - it's better to add eta there
2)  Line 199: better to mention explicitly what "problem parameters".
3) Algorithm 1: it's hard to understand what's phi, psi without reading the text. Please add some high-level names and add some minimal text in line 2, and references to where they're defined in the paper. It might be also a good opportunity to connect Algorithm 3 to Algorithm 1.
4) Is "bistochastic" common? why not doubly stochastic?
5) Since theorem 3 discusses regret, its definition should appear before it and not in Section 3.
6) The fact that the main result of this paper is both the informal Theorem 1 and Theorem 5 is confusing, better to call them both Theorem 1.
7) Line 286 - extra space after "below".
8) The proof of Lemma 3 uses the definition of P_j that only appears in Lemma 2.
9) Lemma 6: strictly speaking, "no policy regret" was never defined, and indeed the phrasing in Proposition 1 is different.
9) Line 569: change doesn't to "does not"
10) Line 587: "equation Equation (16)"
11) Line 605: "actually" is awkward
12) Line 636: note that. Also, "It comes" is nonstandard. Use "It follows" or something else (repeated in several places).
13) Missing periods after the equations in lines 645 and 646.
14) Line 652: If the last term is zero, then it's not ignored - please rephrase.
15) Lemma 11/Line 699: why do you need to show that tao is bounded with probability 1?
16) Line 712: Although I wish "sarcastically dominated" was a thing, that's a typo (:


**Time Spent Reviewing:**

8

---

> ### Author Response · Authors · 2021-08-09
> **Ordered list of response**
>
> Let us first thank you for the time you spent on the paper and your very constructive remarks. All your remarks are on point and should really help us enhance the paper.
> 1. This structure where packets are sent back and forth to servers is a simplified (but equivalent) description of the following interaction: a queue tries to clear a packet through a server. If it was cleared, the queue receives a success message from the server and deletes the packet from its pile. If it was not cleared, the queue receives a failure message and keeps the packet.
>
>     A stronger model taking into account collision would indeed seem more difficult as it would decrease the number of packets cleared at a given time step. However, it would actually improve the performance of the ADEQUA algorithm, simplifying the step where the queues learn each other’s parameters by fixing the feedback’s asymmetry.
>
>     You are right about line 102-104, a queue can (and not has to) send a packet if it is not empty, thank you for pointing this out.
>
>    We did not understand your second question “is there any simple distributed mechanism that implements that without actually submitting the packets?”, but would happily go back at it if our previous remarks did not clarify this point.
>
> 2. In two previous works, it was shown (1) that no policy regret strategies did not guarantee stability when the slack was smaller than $2$, (2) that any Nash equilibria of the patient queuing game was stable when the slack was larger than $e/(e-1)<2$ .
> It therefore  seemed like a natural direction to study whether or not having the queues follow a “patient” version of the regret could give stability guarantees with a slack smaller than 2.
>
>    However, we do agree that the counter-example merely proves that the no-policy regret criteria is not enough by itself, not that *any* no-policy regret strategy would be unstable with a slack smaller than 2. As a matter of fact, queues following algorithm ADEQUA have no-policy regret.
>
>    We still thought it was an interesting point, as it linked our work and its “multi-armed bandits” approach to previous works on the subject, maybe more game theory centered.
>
>    However, if you or other reviewers feel it would improve the paper, we are of course open to moving this section further down or to the appendix.
>
>    a) The two counter-examples are quite different. The one used to prove Theorem 3 sets specific servers and arrival rates, but makes no assumption on the queues behavior other than that they follow “suitable” no external regret policies. In our counter-example, we show that a specific behavior is a “suitable” no policy regret strategy, but leads to an unstable system.
>
>    b) In Theorem 3, the authors proved that when the queues follow “suitable” no external regret policy, the system is unstable if the slack is smaller than 2. Here, we used the same definition of “suitable policy” switching the external regret for the no-policy regret. The goal was to have the two results readily comparable, with no hidden assumption.
>
>   3. We address the common randomness assumption in the response to reviewer fVk3, who also singled out this point.
>
>   4. A non-smooth mapping may fail if two or more rates are equal. However, with the additional assumption that all rates are different, you are right that the usual computation should succeed as soon as the estimation error is small enough. With such a method, we expect the regret bound to also depend on the gap between the two closest rates. We preferred to avoid this dependency with the use of our dominant mapping. It could indeed be an interesting direction to explore.
>
> 5. A conclusion section will be added to the camera ready. Going back to point 4, the dominant mapping may be changed and improved depending on the assumptions.

---

> > ### Comment · Reviewer_E9jT · 2021-08-12
> > **Thanks for the detailed response**
> >
> > I thank the authors for their detailed response. It clarifies most of the issues, and the paper is clear and strong, to begin with. For the sake of perfectionism, I have the following comments:
> >
> > 1. The structure you're describing is indeed very common and natural. Since the way the model is described here deviates from this common structure, I wondered if there's anything substantial behind it. For example:
> >
> > a) Line 105 says that the server "attempts to clear the oldest packet it has received". This implies that the packets were sent and are successfully received, and then the server does this quick check about their age. Then, if the server fails to clear the packet, they'll be sent again even though the server has them.
> >
> > b) In your response, you say that the queue keeps the packets that were not cleared from the server. Why? Did the server discard them? why would it?
> >
> > c) When in real life will the server fail to clear a packet?  isn't that more likely that the packet isn't received, or that the CRC is wrong? These kinds of failures are independent of $\mu$.
> >
> > I think that the more common way to describe this is to say that the server has some random service time for each packet, and never "fails" at anything. I realize these issues are more about the "story" than the actual model assumptions, but to clear any doubts, it might be helpful to rewrite this story in terms that are closer to practice. That is if there is no substantial reason for this deviation from the typical story.  ​The current way the model is written raises unnecessary confusion.
> >
> > 2. I still think this part is not as interesting but I respect the authors' choice, which they justified well.
> >
> > 3. Thanks. It can also help to explain that all randomness here is discrete, so agreeing on a seed is enough even in theory.
> >
> > 4. This is exactly the point, that a complete analysis will depend on the gaps, and then the modification wouldn't be needed anymore. I'm not sure if you really avoided the issue of the gaps by modifying the dominant mapping or that it's just hiding now in Remark 3. The error of the approximate scheme will depend on the gaps, but because it's assumed that $\phi$ can be exactly computed, this will never show up. Logically this makes sense to me, but this logic may hide much since there is a "good reason" why solving a non-smooth strongly convex minimization problem cannot be exact. Giving some evidence or intuition about why this is not the case can make this modification better justified.

---

> > > ### Author Response · Authors · 2021-08-25
> > > **Answer to Reviewer E9jT**
> > >
> > > Thank you again for your detailed comments.
> > >
> > > 1. Indeed, a practical justification for this model might be that the server has random service times for each packet. We will discuss it in the revised version.
> > >
> > >
> > > 4. For computing $\phi$, the objective function is $\frac{1}{K}$-strongly convex and $(\frac{\sqrt{e}}{\hat{\Delta}}+1)$ Lipschitz as explained in Appendix B. It is known that the projected gradient descent converges at a rate $\mathcal{O}\left(\frac{K}{\hat{\Delta}^2 t}\right)$ in this case (see eg Theorem 3.9 from Convex Optimization: Algorithms and
> > > Complexity by Bubeck). The estimation error of $\phi$ coming from the computation thus only scales with $\Delta$ (and $K$), but is independent from the gaps in $\lambda$ and $\mu$. Here again, a small detail is hidden as the projection can only be (quickly) approximated using quadratic programming. For similar reasons, the convergence rate of the involved quadratic program does not depend on the gaps between $\mu$ and $\lambda$.
> > > Moreover, our simulations illustrate that we can efficiently approximate $\phi$ (up to a small enough error), although the minimal gap between $\lambda$ and $\mu$ is $0$ in the considered instances. Because of this, our algo should be unstable without the regularization term used in $\phi$.

---

### Official Review · Reviewer_Vdob · 2021-07-16

**Rating:** 5
**Confidence:** 1

**Summary:**

This paper studies a decentralized learning problem for queueing systems. In my opinion, the paper is difficult to follow. I could not understand the problem setup and the paper’s contributions. Also, the article has a strong economics component which I am not familiar with.

**Limitations And Societal Impact:**

Yes

**Main Review:**

From my understanding of the model in Sec. 2, and the cooperative algorithm in Sec. 3, it seems that the problem addressed is closely connected to scheduling/routing problems studied extensively in computer networks. See, for example,

Georgiadis, L., Neely, M.J. and Tassiulas, L., 2006. Resource allocation and cross-layer control in wireless networks. Now Publishers Inc.

It would be helpful if the authors can clarify why their problem cannot be solved with mainstream networking approaches.

In section 4.2, the authors propose an “ordered Birkhoff” algorithm, which seems to be a variation of the original Birkhoff algorithm. Many works in the literature study variants of the Birkhoff algorithm, but none of them are mentioned in the paper. See, for example,

Dufossé, F. and Uçar, B., 2016. Notes on Birkhoff–von Neumann decomposition of doubly stochastic matrices. Linear Algebra and its Applications, 497, pp.108-115.

How is your “ordered Birkhoff” algorithm different from existing algorithms?

**Time Spent Reviewing:**

4

---

> ### Author Response · Authors · 2021-08-10
> **Ordered Birkhoff and link with scheduling/routing problems**
>
> Thank you for your review.
>
> Previous variants of the Birkhoff algorithm are non-smooth in the input parameters. This is problematic as around discontinuities, queues with arbitrarily close estimates could get very different output matrices and collide a large number of times in the exploitation phase. The ordered Birkhoff algorithm is smooth in the input parameters and fixes that issue. However, you are right that a reference to previous algorithms and a discussion of that point is missing in the paper, thank you for pointing that out.
>
> In the provided reference, we didn’t find mentions of fully decentralized learning, which is one of the major difficulties of the problem treated in the paper.
> However, there still seems to be a relation between the two problems, and it would be very interesting to study it further, thank you for the reference.

---

> > ### Comment · Reviewer_Vdob · 2021-09-02
> > **Thanks for the author's response.**
> >
> > Thanks for the author's response.
> >
> > After reading the other reviews, I'll raise my score to 5.
> >
> > I'd like to make three suggestions:
> >
> > 1) I believe the $K^2$ bound in Lemma 2 can be sharpened to $(K-1)^2+1$. The sharper bound follows by Carath\’edory's theorem since the Birkhoff polytope is embedded in an $(N-1)^2$-dimensional subspace.
> >
> > 2) In the numerical results, the authors show how the new algorithm can stabilize a system of queues when $\eta < 2$. However, the paper claims that the new algorithm stabilizes any system when $\eta >1$. I suggest including a figure in the appendix that shows the average queue occupancy as $\eta \to 1$.
> >
> > 3) Algorithm 3 could terminate earlier if the permutation matrices cover the traffic demand [Equation in line 271]. For future work, it would be interesting to make the $K^2$ bound in Lemma 2 depends on the system "slackness."

---

### Official Review · Reviewer_fVk3 · 2021-07-17

**Rating:** 8
**Confidence:** 4

**Summary:**

The paper considers an online queueing system where multiple queues
  are served by a set of servers. Each queue (if non-empty) sends a
  packet to one of the servers. Each server serves the oldest packet
  it receives, and returns the other packets to their respective
  queues. The goal is to design a policy so that the queues remain
  stable. Past work has shown that the stability of the system depends
  on its /slack/, which is a measure that captures the ratio between
  total service rate of a subset of servers and the total arrival rate
  to a subset of queues. Specifically, it is known that if the slack
  is strictly greater than one, then there exists centralized policies
  that are (strongly) stable. On the other hand, restricting attention
  to policies that are decentralized across queues, stable policies
  were only known to exist when slack was strictly greater than two
  (or greater than $e/(e-1)$ if the arrival rates and service rates
  were known).

  In this work, the authors propose a decentralized policy that
  achieves stability for all values of slack strictly greater than
  one, matching the stability regime for centralized policies. The
  algorithm uses shared randomness to coordinate the queues' actions
  to explore the arrival rates and the service rates, and then use the
  estimates in the exploitation phase. The challenge is to ensure that
  the queues avoid colliding on the same server; this is done using
  smooth dominant mappings (which take the estimates and return a
  bistochastic fractional matching between servers and queues) and
  regular Birkhoff-von Neumann (BvN) decompositions (which take the
  fractional bistochastic matching and yield a distribution over a
  matching). The authors support their theoretical results through
  numerical computations that show the performance of their proposed
  algorithm.

**Limitations And Societal Impact:**

Yes

**Main Review:**

*  Pros
   - The paper studies the setting of decentralized control with
     carryover effects, and shows that decentralized policies with
     shared randomness can be as effective as centralized policies in
     acheiving stability. This is interesting from the theoretical
     perspective, characterizing the strength of decentralized
     policies.
   - The algorithm uses smoothed/regularized versions of the dominant
     matchings and BvN decompositions to ensure continuity of the
     matching with respect to the estimates. This might be of interest
     more broadly.
* Cons
   - The strong stability bounds are super-exponential in the number
     $K$ of servers, leading to impractically large coefficients even
     for $K$ as small as $5$. Since the shared numerics seems to
     suggest good performance of the algorithm, it would suggest that
     the theoretical analysis is not tight.
   - The algorithm requires a substantial amount of shared randomness,
     which may not be possible in practical settings without a
     substantial level of centralization.

Overall, I think the paper makes a valuable theoretical contribution
to the problem of decentralized control of queueing systems (or more
generally systems with carryover effects).

**Time Spent Reviewing:**

5

---

> ### Author Response · Authors · 2021-08-09
> **About the shared randomness**
>
> Let us first thank you for the time you spent reading the article and for your feedback.
>
> On the subject of the shared randomness, some level of coordination between the queues is indeed assumed in the model. First, the queues have a common time discretization (although they are not always active). This is often referred to as synchronicity in decentralized systems. Secondly, all the queues start at the same time t=1, which is referred to as the static assumption in multiplayer bandits. These two assumptions are actually what makes the shared randomness equivalent to the use of a common seed. Studying queuing systems with neither synchronicity or static assumption is left open for future work and presents a significant challenge, as in most decentralized systems (eg multiplayer bandits).

---

> > ### Comment · Reviewer_fVk3 · 2021-09-13
> > **Re**
> >
> > Thanks for the clarification about shared randomness. My evaluations remain the same.

---

### Official Review · Reviewer_7uFN · 2021-07-18

**Rating:** 7
**Confidence:** 4

**Summary:**

For the recently proposed online queueing system, the authors presented several key results. First, they demonstrated that, when the slack is below 2, coorperation is necessary for the stability of the system by constructing a counterexample; second, they proposed a decentralized learning algorithm, and proved that it guarantees stability under the mild condition of the slack being larger than one.

**Ethics Review Area:**

["I don’t know"]

**Limitations And Societal Impact:**

not apply

**Main Review:**

The paper contains important new results and (counter)examples for online queueing system. The decentralized learning algorithm (ADEQUA) is an important contribution to this area of research, I could see it being further analyzed by many researchers in the future, and variations being proposed. The counterexample (in Proposition 1) and the main results (Theorem 5) fill in the gap of knowledge of this important problem.



**Time Spent Reviewing:**

three

---

> ### Comment · Reviewer_7uFN · 2021-09-10
> **Thanks to the authors' response**
>
> Appreciate the authors' detailed response, and my evaluation remains the same.

---

### Decision · Program_Chairs · 2021-09-27

**Decision:**

Accept (Spotlight)

**Comment:**

The paper makes a clear contribution to the question of decentralized learning in queuing systems.
Thus, we decided to accept the paper.